# The effect of interactive ozone chemistry on weak and strong stratospheric polar vortex events

Jessica Oehrlein[1], Gabriel Chiodo[1,2], and Lorenzo M. Polvani[1]

[1]Department of Applied Physics & Applied Mathematics, Columbia University, New York, United States
[2]Department of Environmental Systems Science, Institute for Atmospheric and Climate Science, ETH Zurich, Switzerland

**Correspondence:** Jessica Oehrlein (jessica.oehrlein@columbia.edu)

**Abstract.** Modeling and observational studies have reported effects of stratospheric ozone extremes on Northern Hemisphere spring climate. Recent work has further suggested that the coupling of ozone chemistry and dynamics amplifies the surface response to midwinter sudden stratospheric warmings (SSWs). Here, we study the importance of interactive ozone chemistry in representing the stratospheric polar vortex and Northern Hemisphere winter surface climate variability. We contrast two simulations from the interactive and specified chemistry (and thus ozone) versions of the Whole Atmosphere Community Climate Model, designed to isolate the impact of interactive ozone on polar vortex variability. In particular, we analyze the response with and without interactive chemistry to midwinter SSWs, March SSWs, and strong polar vortex events (SPVs). With interactive chemistry, the stratospheric polar vortex is stronger, and more SPVs occur, but we find little effect on the frequency of midwinter SSWs. At the surface, interactive chemistry results in a pattern resembling a more negative North Atlantic Oscillation following midwinter SSWs, but with little impact on the surface signatures of late winter SSWs and SPVs. These results suggest that including interactive ozone chemistry is important for representing North Atlantic and European winter climate variability.

## 1 Introduction

The climate impacts of stratospheric ozone extremes, particularly Antarctic ozone depletion, have been widely studied (Previdi and Polvani (2014) and references therein). While the effects are clearer and larger in the Southern Hemisphere, ozone extremes have also been shown to be associated with springtime surface anomalies in the Northern Hemisphere (Smith and Polvani, 2014; Calvo et al., 2015; Ivy et al., 2017).

Polar cap ozone anomalies are strongly related to interannual variability in stratospheric polar vortex strength, which is larger in the Northern Hemisphere than the Southern Hemisphere. This is a result of the larger amplitudes of upward-propagating planetary waves, which perturb the stratospheric circulation. Years with low wave activity tend to correspond to a stronger vortex and a weaker Brewer-Dobson Circulation (BDC), resulting in weaker ozone transport from the tropics into the poles and decreased mixing across the vortex edge, as well as enhanced formation of polar stratospheric clouds, which contribute to increased springtime destruction of ozone. Years with high wave activity correspond to a weaker vortex and a stronger BDC, with stronger ozone transport from the tropics and increased mixing (Newman et al., 2001).

These processes are well-represented in fully interactive chemistry-climate models (Strahan and Douglass, 2004). However, such models are computationally expensive compared to the more common ones, in which stratospheric ozone is simply prescribed. A number of studies have explored the importance of interactive ozone chemistry on model representations of coupled stratosphere-troposphere variability. Smith and Polvani (2014) and Karpechko et al. (2014) found little impact of stratospheric ozone extremes on surface climate in the Northern Hemisphere using prescribed zonal mean monthly mean

ozone fields. However, Calvo et al. (2015) found robust surface impacts associated with stratospheric ozone extremes using an interactive chemistry-climate model, suggesting the potential importance of this coupling. Further model studies are needed to disentangle the effects of ozone from those of polar vortex variability.

While the effect of polar stratospheric clouds on ozone is mainly seen in the spring when sunlight returns to the region, the variability of the polar vortex can result in wintertime ozone anomalies which may have surface impacts. The most extreme

states of the polar vortex are sudden stratospheric warmings (SSWs) and strong polar vortex events (SPVs). We define these precisely in Section 2, based on extreme values of zonal mean zonal wind. Leading up to an SSW, dynamical forcing disrupts the stratospheric circulation, eventually resulting in a reversal of zonal mean zonal wind throughout much of the polar strato-sphere. SSWs have surface effects for the two months following, particularly a negative North Atlantic Oscillation and cold anomalies over much of Northern Eurasia. Conversely, SPVs, in which abnormally strong westerly zonal mean zonal winds

occur, are the result of anomalously weak planetary wave activity over a protracted period. As such, they are not rapid dynam-ical events in the same way as SSWs, but they may still have surface impacts, which is typically a positive NAO (Baldwin and Dunkerton, 2001).

For the dynamical reasons described above, SSWs and SPVs tend to be associated with the occurrence of positive and negative stratospheric ozone anomalies, respectively. About two weeks prior to an SSW, the BDC accelerates, resulting in

adiabatic warming of the stratosphere and enhanced isentropic eddy transport of ozone and thus increased ozone concentration over the pole (De La Cámara et al., 2018). SPVs are similarly accompanied by an anomalously weak BDC because of the lack of planetary wave activity and thus an anomalously low transport of ozone, as well.

Because they affect both stratospheric ozone and the NAO in the troposphere, extreme vortex events offer an ideal case in which to study wintertime surface impacts of ozone chemistry. Haase and Matthes (2019) studied the impact of interactive

versus prescribed ozone on SSWs, as well as their surface effects, in simulations of the recent past (1955-present) in an earth system model. They compared results of a simulation with interactive ozone to those of a simulation with prescribed ozone. This prescribed ozone was given daily (with no averaging/climatology) from a single historical interactive chemistry simulation. They found a stronger climatological vortex in the interactive chemistry simulation, and this was associated with a decreased SSW frequency. Further, SSWs were followed by stronger and more persistent surface anomalies in the simulation

with interactive chemistry. These results suggest important surface impacts of ozone chemistry. However, their simulation was relatively short (64 winters), and the historical period they simulated includes long-term trends in ozone that may affect the results. Also, their method of prescribing ozone means that the ozone in the specified chemistry simulation was associated with dynamical variability of the interactive chemistry run, and that variability was inconsistent with the dynamical state of the specified chemistry model.

Building on the study of Haase and Matthes (2019), we here study interactions between ozone chemistry and polar vortex variability by analyzing SSWs, SPVs, and their surface impacts in two 200-year timeslice simulations with fully interactive and prescribed chemistry versions of a model. Using 200-year timeslices provides us with a large sample size of SSW and SPV events without long-term ozone trends, and we prescribe ozone based on the ozone climatology from the 200 years of the interactive chemistry simulation. Due to the larger sample size, climatological ozone distribution, and constant forcings,

this set of simulations more clearly separates the impact of ozone's interannual variation on stratosphere-troposphere coupling. While we do not see decreased SSW frequency with interactive chemistry, we confirm Haase and Matthes (2019)'s results on the vortex climatology and response to midwinter SSWs. We further find that there is little surface effect of interactive ozone chemistry immediately following SPVs or March SSWs. However, SPVs show long-lasting effects on stratospheric ozone, with anomalies 1-2 months after the central date of a similar magnitude to those caused by midwinter SSWs.

The paper is organized as follows. Section 2 describes the model, simulations, and methodologies. Section 3 addresses our results on the impacts of interactive chemistry, considering the stratospheric mean state, midwinter SSWs, March SSWs, and SPVs. We conclude the paper with a discussion of these results.

## 2    Methods

In this study, we analyze model integrations performed with the Whole Atmosphere Community Climate Model, Version 4

(WACCM4), one atmospheric component of the Community Earth System Model (CESM1) (Marsh et al., 2013). WACCM4 is an interactive chemistry-climate model with a horizontal resolution of $1.9°$ in latitude and $2.6°$ in longitude, 66 vertical levels, and a model top at 5.1 $\times10^{-6}$ hPa (140 km). Northern Hemisphere stratospheric variability, such as the frequency and dynamical features of SSWs, are accurately simulated in WACCM4 (Marsh et al., 2013).

We perform two model integrations, both 200-year-long timeslice integrations with forcings at constant year-2000 values to

avoid long-term trends in ozone. One model integration uses the fully interactive chemistry scheme in WACCM4 (Kinnison et al., 2007). We refer to this simulation as the CHEM simulation in the analysis. The other uses the "Specified Chemistry" version of WACCM, known as SC-WACCM (Smith et al., 2014). We refer to this prescribed chemistry simulation as the NOCHEM simulation in the analysis. In the NOCHEM simulation, ozone concentrations (and other radiatively active atmospheric constituents, including CFCs) are prescribed using zonally symmetric, monthly mean, seasonal climatology computed

from the WACCM integration. These zonally symmetric monthly ozone fields are read into SC-WACCM and interpolated linearly to the day of the year. More details can be found in Smith et al. (2014). Hence, both CHEM and NOCHEM strictly impose identical year 2000 forcings for all radiatively active species, and only differ in their treatment of ozone. The use of climatological ozone fields in NOCHEM removes the effect of extreme ozone variations on the climate system. One might consiser specifiying non-zonally symmetric ozone (Haase and Matthes, 2019), but that comes at cost of a major physical inco-

sistency between the polar vortex and the ozone field: in other words the exteme ozone years in the model will not correspond with the unperturbed vortex years. More importantly, the vast majority of climate models in CMIP specificy zonally symmetric

stratospheric ozone, including within CMIP6 (Keeble et al., 2020): hence the zonally-symmetric specified ozone case is the one of most interest in terms of evaluating the impact of interactive ozone chemistry.

We identify SSWs in the model output following the definition in Charlton and Polvani (2007a) (see the corrigendum Charlton-Perez and Polvani (2011)). We define an SSW as a reversal of zonal mean zonal wind at $60°$ N and 10 hPa from westerly to easterly during November through March, with the central date being the first day of easterly zonal mean zonal winds. No later date can be a central date until the winds have been westerly again for at least 20 days, and the winds must return to westerly for at least 10 consecutive days before April 30 (thus discarding stratospheric final warmings). This definition is optimal for identifying SSWs, as described by Butler and Gerber (2018). We focus on SSWs occurring in December-February and in March. We consider March events separately from December-February events due to different shortwave heating behavior, model bias in March SSW frequency (too frequent SSWs in our model), and different NAO structure in early spring compared to winter.

To the best of our knowledge, there is no standard definition of an SPV. Different methods have been used in the literature (Limpasuvan et al., 2004; Tripathi et al., 2015; Scaife et al., 2016; Beerli and Grams, 2019). We here follow the definition used in Scaife et al. (2016) and Smith et al. (2018), designed to be analagous to the Charlton and Polvani (2007a) SSW definition and to result in a similar number of events in reanalysis. We define an SPV as zonal mean zonal wind at $60°$ N and 10 hPa reaching 48 m/s or higher (westerly) during November through March, with the central date being the first day of zonal mean zonal winds above 48 m/s. No later date can be a central date until the winds return below 48 m/s for at least 20 consecutive days. We focus on SPVs occurring in December-February, due to low event frequency in November and March. A separate analysis reveals that results are not sensitive to using a 41.2 m/s threshold as in Tripathi et al. (2015).

The results we present here are based on composites of daily model output for climate variables, with composites centered around SSW or SPV central dates. For composites from either CHEM or NOCHEM simulations, we calculate significance using a Monte Carlo test based on 5000 randomly chosen central dates. We also consider the difference in CHEM or NOCHEM composites, denoted CHEM-NOCHEM; for these, we calculate significance from a two-sided two-sample t-test.

## 3 Impact of interactive chemistry

### 3.1 Stratospheric mean state and extreme events

We first consider the effect of interactive chemistry on the mean state of the stratosphere by examining the climatological Northern Hemisphere 10 hPa zonal mean zonal wind (Figure 1). We find stronger westerlies in CHEM than in NOCHEM in the vortex formation stage (September and early October) and in the latter half of winter (January-April), between $60° - 80°$ N. In line with this, we also find weaker downwelling in winter in the upper latitudes in CHEM than in NOCHEM (not shown). This relative strength in CHEM in late winter also corresponds to a delayed final warming by 7 days on average. These results are in agreement with those of Haase and Matthes (2019). We also found similar results in six 1955-2005 historical integrations of WACCM and SC-WACCM (with ozone specified monthly or daily from the WACCM climatology) from Neely et al. (2014) (not shown), further indicating that this feature is robust.

This is not the case in Smith et al. (2014), where the vortex is of similar strength with interactive and prescribed ozone under constant year 1850 conditions. The difference between that study and ours is the level of chlorofluorocarbons (CFCs): these are zero in Smith et al. (2014), which simulates pre-industrial conditions, but they are substantial in our study, which simulates year 2000 conditions. They are similarly substantial in the historical (1955-present day) simulations in Haase and Matthes (2019) and Neely et al. (2014). Because the differences between interactive and specified ozone simulations depend on the level of CFCs, a precise understanding of the mechanisms for the difference will require disentangling the dynamics and chemistry. Higher ozone variability in the presence of CFCs (Calvo et al., 2015) might increase the effects of the ozone-dynamics feedbacks, rendering this a very difficult problem. There are indications that these differences may be related to zonal asymmetry of ozone (Haase and Matthes, 2019), further complicating the relationship. Albers and Nathan (2012) have proposed a complex mechanism to detail the coupling of zonally asymmetric ozone and dynamics in the context of a highly idealized linear model. In their model, zonal asymmetries in ozone precondition the waves, causing a reduction in planetary wave drag and a colder polar vortex. However, determining whether this mechanism is operative in our comprehensive model would be quite difficult, as the mechanism relies on many assumptions that are likely inapplicable in the presence of highly nonlinear, time-dependent, breaking waves as are observed in the winter polar stratosphere in a fully interactive model.

Because we identify extreme stratospheric events using zonal mean zonal winds at 10 hPa and 60° N (U1060) (Charlton and Polvani, 2007a; Butler and Gerber, 2018), we next examine the mean state and variability of this quantity in CHEM and NOCHEM. Figure 2 shows the two distributions of U1060 in December through March. The average difference in DJFM between CHEM and NOCHEM is about 1.7 m/s. To determine whether this is statistically significant, we consider the average zonal mean zonal winds over each winter and treat the winters as independent. A two-sample, two-tailed Welsh's t-test of DJFM average winds in CHEM and NOCHEM yields a p-value of 0.023, so the difference, though small, is significant at a 95% level. The CHEM distribution also has a longer right tail, which is consistent with the polar vortex being stronger overall with interactive chemistry. It also indicates that we should expect more SPVs in CHEM than in NOCHEM. While there are fewer days of weak westerlies (0-20 m/s) in CHEM than in NOCHEM, the numbers of days of easterlies are similar, so we expect less of a difference in SSW frequency between the two simulations.

Indeed, this is what we find when we calculate the frequencies of weak and strong vortex events in the CHEM and NOCHEM simulations (Table 1). We consider December-February (DJF, midwinter) and March (late winter) separately for two reasons. First, the ozone impacts in midwinter are different from those in late winter/early spring, as shortwave effects become important in spring. Second, our model is biased in March, with too many SSWs compared to reanalysis, a feature also seen in more recent versions of this model (Gettelman et al., 2019). We see 1.4 March SSWs per decade in NOCHEM and 1.95 March SSWs per decade in CHEM compared to 0.87-1.1 per decade in the reanalysis (Butler et al., 2017).

The stronger vortex in midwinter in the CHEM simulation might lead us to expect fewer DJF SSWs in CHEM than in NOCHEM. We do see a decrease of about 10% in DJF SSWs with interactive chemistry compared to specified chemistry, but this decrease is far from being statistically significant. In contrast, in March, we see more SSWs in CHEM than in NOCHEM, potentially related to the later breakdown of the vortex.

Haase and Matthes (2019) consider the overall (November-March) number of SSWs. They report a decrease in overall SSWs with interactive chemistry of around 30%. In contrast, for November-March, we find virtually no difference in SSWs (109 events vs 111, not shown) in CHEM and NOCHEM. We note that both of these frequencies, around 5.5 events per decade, are on the lower end of what is seen across reanalyses (Butler et al., 2017; Cao et al., 2019) but very well within the spread among state-of-the-art chemistry-climate models (Ayarzagüena et al., 2018).

We now consider SPV frequency. The increase in DJF SPV frequency from NOCHEM to CHEM is about 29%. This is unsurprising given the stronger vortex in CHEM overall. With our definition of SPVs, the number of March strong vortex events (in either simulation) is too small for a robust statistical analysis. This is because of the weaker vortex in March compared to DJF; a much larger anomalous vortex strength would be necessary to reach 48 m/s. Because of the low number of such events, we do not further study March SPVs and thus discard them from the analysis.

We now examine DJF SSWs, March SSWs, and DJF SPVs separately in each of the following three sections.

## 3.2 Midwinter sudden stratospheric warmings

We start by focusing on the surface impacts of SSWs, seeking to document any differences between the CHEM and NOCHEM simulations. After noting the impact of the events on the surface, we then consider how any differences in those impacts arise aloft.

Figure 3 shows composite surface level pressure anomalies in the first and second months (top and bottom respectively) following December-February SSWs in CHEM (left, 75 events) and NOCHEM (middle, 67 events), as well as the difference between the two (right). We see a strong and significant pattern resembling a negative North Atlantic Oscillation (NAO) in the first month following SSWs in both CHEM and NOCHEM, and in both cases this negative annular mode persists through the second month following the event. There is minimal difference between the two simulations in the first 30 days, with the CHEM simulation having only a slightly stronger signal. However, the difference is statistically significant and strongly projects onto the NAO 30-60 days after the central date. This indicates that the surface signature of SSWs is stronger and more persistent in CHEM than in NOCHEM.

To determine whether the differences at the surface following SSWs in CHEM and NOCHEM are a result of differences originating in the stratosphere, we calculate the Northern Annular Mode (NAM) for each simulation. We use a method similar to that of Gerber et al. (2010) and Gerber and Martineau (2018); the detailed procedure is in Appendix A. We show the results of the NAM calculations in Figure 4. The CHEM and NOCHEM composites around SSWs have comparable NAM anomalies in the stratosphere around the central date, but in the CHEM simulation the negative anomaly persists more strongly in the lower stratosphere beyond 40 days after the central date. The CHEM-NOCHEM difference shows that this change in persistence with interactive chemistry is significant at the 95% level. There is also more descent of the anomaly to the surface in the CHEM simulation, especially at about 30 days after the central date.

This difference in descent is also seen in the CHEM-NOCHEM temperature anomalies (Figure 5a). The warming in the stratosphere associated with the onset of the SSW is larger with interactive chemistry. This stratospheric temperature anomaly

then descends more strongly through the stratosphere and troposphere in the CHEM simulation than in the NOCHEM simulation.

We investigate the processes leading to these changes in more detail by examining the dynamical, longwave, and shortwave
heating terms. The greater warming throughout the stratosphere is due to increased dynamical heating (Figure 5b) in CHEM compared to NOCHEM, as the higher temperature with interactive chemistry is also associated with a longwave cooling response (Figure 5c). The higher stratospheric temperatures result in greater longwave emission. The increase in dynamical forcing also corresponds to increased ozone transport. Ozone is a longwave emitter, so the increased dynamical forcing could directly account for part of this longwave cooling difference, as well.

The increased dynamical heating in CHEM could be related to greater wave activity necessary for an SSW to occur with a stronger mean vortex state. Figure 6 shows the eddy heat flux over 40-80° N over time in CHEM and NOCHEM. This is stronger by about 2 mK/s around the central date in CHEM than in NOCHEM, indicating a slightly stronger wave forcing in CHEM. The CHEM and NOCHEM means are at the upper and lower bounds of the other's confidence intervals, respectively. Further, the zonal mean zonal winds at 10 hPa and 60° N around the central date of the SSW (shown in Figure 7) are both
stronger prior to the event and more easterly following the central date in CHEM than in NOCHEM. However, the residual vertical velocity anomalies leading up to SSWs is nearly identical for CHEM and NOCHEM (not shown), so the increased dynamical heating in CHEM might be a result of a stronger vertical temperature gradient related to the stronger vortex in this simulation (associated with a colder polar stratosphere).

In DJF, the dynamical heating and the longwave heating are the dominant temperature tendency terms. There is also a
210 significant shortwave heating response (Figure 5d), but in midwinter it is an order of magnitude smaller than the other terms, owing to the absence of incoming solar radiation to polar night. The structure in height and time is related to integrated effects of the ozone anomalies following the SSW, which show a similar structure (Kiesewetter et al., 2010). The importance of the shortwave response increases the later in winter the SSW events occurs. We illustrate this in Figure S1, showing much stronger differences in CHEM and NOCHEM shortwave anomalies for February SSWs than for December or January events.

Finally, we examine the anomaly in total ozone column around the central date of the SSW (Figure 7) in the CHEM simulation. We see a sharp increase in ozone in the 15 days leading up to the central date, reaching a peak of on average about 40 Dobson units above climatology just after the central date, similar to that seen in reanalysis and a similar model by De La Cámara et al. (2018). This ozone anomaly results from transport due to the greater dynamical forcing in CHEM, as noted earlier. Following the central date, anomalies of about 20 Dobson units persist for up to 3 months following the central date. This
ozone anomaly is consistent with total ozone column in reanalysis and a similar model (De La Cámara et al., 2018) and the smaller ozone depletion in years with early SSWs observed by Strahan et al. (2016).

In summary, DJF SSWs are preceded by larger wave forcing in CHEM than in NOCHEM, partially because of the stronger mean state of the polar vortex. This then results, on average, in more intense SSWs, stronger stratosphere-troposphere coupling, a more negative NAO-like pattern at the surface, and long-lasting stratospheric ozone anomalies.

## 3.3 March sudden stratospheric warmings

We now turn to the March SSWs. Figure 8 shows the composite sea level pressure anomalies for CHEM and NOCHEM, as well as the CHEM-NOCHEM difference, for each of the first two months following the central date. Both simulations again show a negative NAO-like pattern in the two months following the SSW. There are some regions with significant difference between CHEM and NOCHEM in the first thirty days, but the pattern does not project strongly onto the NAO. Also, there is very little difference between the two composites in the second thirty days after the central date.

The surface responses seen following March SSWs, in both models, are weaker and less persistent than those following DJF SSWs, and the areas of strong or significant low or high anomalies are smaller. Three factors could contribute to this: weaker SSWs, weaker stratosphere-troposphere coupling, and a shorter NAM decorrelation timescale in March than in DJF (Baldwin et al., 2003; Simpson et al., 2011), resulting in weaker anomalies at the surface when averaged over several weeks. The differences between surface impacts of SSWs in CHEM and NOCHEM are also weaker for March SSWs. Thus, interactive ozone seems much less important for the surface effects of March SSWs than for DJF SSWs.

Considering the NAM in these simulations as shown in Figure S2, we see negative NAM anomalies at the surface in both the CHEM and NOCHEM simulations, consistent with the negative NAO-like pattern seen in the Figure 8. There is a stronger signal in the troposphere in the CHEM compared to NOCHEM March SSW simulations at around 15-20 days after the central date, which may correspond to the surface pressure differences.

The NAM anomalies suggest that March SSWs in both CHEM and NOCHEM are weaker overall than the DJF SSWs; the stratospheric NAM anomalies are smaller and less significant. The eddy heat flux show in Figure S3, however, shows weaker wave forcing preceding only the CHEM (not the NOCHEM) March SSWs compared to those in DJF. Stratosphere-troposphere coupling also seems weaker compared to that seen for DJF SSWs. Further, the difference in the NAM descent between CHEM and NOCHEM is less strong and persistent than the difference seen after midwinter SSWs.

Soon after the central date for March SSWs, the NAM signal in the stratosphere is weaker with CHEM than NOCHEM, in contrast to the midwinter SSW case. This difference appears to arise from the temperature and heating anomalies (Figure S4). The lower stratosphere is only briefly and weakly warmer in CHEM compared to NOCHEM. Shortwave heating seems to be dominant in the temperature response to March SSWs, with the CHEM-NOCHEM difference in temperature anomalies (Figure S4a) largely following the difference in shortwave heating anomalies (Figure S4d). This is in contrast to the DJF SSWs, where the shortwave heating had little effect, and dynamical heating was dominant.

Finally, we note that unlike the DJF SSW case, the ozone anomaly for March SSWs does not persist after the event (Figure 9). This is related to the seasonal breakdown of the vortex, seen in the wind curves. Because these are late winter SSWs, the second month following the central date is near the expected stratospheric final warming date; the winds return to easterly about 50 days after the March SSW central date. The ozone anomaly returns to 0 Dobson units as the vortex breaks down. The maximum ozone anomaly is also about half the size of the maximum anomaly seen in DJF, consistent with the weaker nature of the March SSW events overall.

### 3.4 Midwinter strong polar vortex events

Finally, we turn our attention to strong polar vortex (SPV) events in DJF. While less extensively studied than SSWs, SPVs also impact surface climate. Baldwin and Dunkerton (2001) suggest that strong polar vortex events can have surface signals comparable to but opposite in sign to those following SSWs, and Smith et al. (2018) found effects of Northern Hemisphere SPVs on spring and summer Arctic sea ice.

In the thirty days following the SPV central date, we see a pattern reminiscent of a weakly positive NAO in both CHEM and NOCHEM (Figure 10). This positive NAO-like pattern appears stronger in CHEM than in NOCHEM, but not significantly so. There is very little difference from climatology at the surface in the second month after the event in either of the simulations. This minimal difference using interactive versus specified ozone compared to the difference seen with SSWs may be related to the more zonal nature of SPVs. We specify ozone in a zonally-symmetric way, which is much more consistent with the vortex seen in an SPV than in an SSW.

The NAM anomalies following SPVs in CHEM and NOCHEM (Figure S5) have similar strength (and opposite sign) in the stratosphere to those following midwinter SSWs, but they have much weaker downward propagation, consistent with an only weakly positive NAO. The difference between the NAM anomalies in CHEM and NOCHEM confirms a more positive NAM in mid-to-lower troposphere in the first month following the SPV central date with interactive chemistry, but again, this difference is not significant and does not reach the surface.

These minimal differences in surface pressure and NAM are consistent with the similarity in the evolution of stratospheric temperature and heating rates in CHEM and NOCHEM, shown in Figure S6. The only large and significant difference is in stratospheric temperature, 40-60 days following the SPV central date, when the stratosphere is colder with interactive chemistry. This is after zonal mean zonal winds have returned to typical levels and is thus likely related to the stronger mean state of the stratospheric polar vortex with interactive chemistry compared to specified chemistry.

The zonal mean zonal winds in CHEM and NOCHEM around the SPV central dates further confirm that there is little difference in the strength of these events between CHEM and NOCHEM; the winds follow nearly identical trajectories from 30 days before to 30 days after the central date. We also see a weaker ozone anomaly following SPVs than following SSWs, with a maximum absolute anomaly of about 30 Dobson units compared to 40 (Figure 11). The ozone decrease following SPVs is also much more gradual than the increase seen in DJF SSWs. This is consistent with the fact that SPVs are not strong and sudden dynamical events in the way that SSWs are. As with DJF SSWs, though, the anomaly does persist for three months after the central date.

### 4 Conclusions

The climate model results presented here show an important relationship between interactive ozone, the climatological state of the stratospheric polar vortex, and the Euro-Atlantic surface impacts of midwinter SSWs. However, ozone chemistry has minimal impact on the surface effects of March SSWs and of midwinter SPVs, despite long-lasting total ozone column anomalies in the latter case. Furthermore, in contrast to the results reported by Haase and Matthes (2019), we do not find significantly

fewer SSWs with interactive chemistry, despite the stronger climatological polar vortex. However, we do find more frequent SPVs.

The stronger polar vortex mean state with interactive ozone chemistry also affects the surface signature of SSWs. A possible mechanism is that stronger wave forcing is necessary for an SSW to occur, and the resulting negative NAM propagates to the surface more strongly, as well. This result is also consistent with that reported by Haase and Matthes (2019), though the effects documented here are weaker. In extending this work to consider March SSWs, we found that while the same stronger dynamical forcing is present, the influence of the shortwave heating term in late winter/early spring results in a stratospheric temperature difference of opposite sign, and with little difference at the surface following March SSWs between interactive chemistry and specified chemistry simulations. We also find minimal impact on midwinter surface effects of SPVs. However, we do see persisting negative ozone anomalies that can have an important effect in spring (Ivy et al., 2017).

Previous work (Smith and Polvani, 2014; Calvo et al., 2015; Ivy et al., 2017; Lin et al., 2017; Rieder et al., 2019) has shown the importance of ozone for the stratospheric polar vortex and surface springtime climate variability. Haase and Matthes (2019) further suggested that feedbacks among chemistry and dynamics are important for accurately capturing the response at the surface to SSWs, one of the major drivers of North Atlantic and European winter climate variability. By running longer simulations allowing for a cleaner quantification of the impact of interactive ozone, we find that these feedbacks are important for representing impacts of midwinter SSWs. However, we do not find similar importance for describing surface response to March SSWs or DJF SPVs. Our results suggest that including interactive ozone chemistry may have a sizable impact on N.Atlantic and European winter and spring climate variability in models.

Finally, we note that while we have only focused on winter SSWs and SPVs, stratospheric final warmings also have tropospheric effects (Black et al., 2006; Ayarzagüena and Serrano, 2009; Wei et al., 2007; Hardiman, 2011; Thieblemont et al., 2019; Butler et al., 2019). Those effects are dependent on the timing of the final warming, with earlier final warmings resulting in surface effects more like those seen following SSWs (Ayarzagüena and Serrano, 2009; Li et al., 2012). Interactive chemistry may thus also affect the representation and surface signature of stratospheric final warmings in models; this will be investigated in a follow-up study.

*Data availability.* All the model output is currently stored at the High Performance Storage System (HPSS) repository at the National Center for Atmospheric Research (NCAR). More specifically, the data can be found under the experiment tags "CO2x1SmidEmin_yBWCN" (CHEM) and "b.e10.B2000WSCCN.f19_g16.control.001" (NOCHEM). Additionally, the data are available from the corresponding author upon request.

*Author contributions.* GC and LMP designed the model experiment. JO, GC, and LMP decided on the analysis and wrote the paper. GC carried out the model simulations. JO performed the data analysis and produced the figures.

*Competing interests.* The authors declare that they have no conflict of interest.

*Acknowledgements.* We thank several anonymous referees for their helpful comments. J. Oehrlein is funded by National Science Foundation (NSF) grant DGE 16-44869. G.Chiodo is funded by the SNF Ambizione grant PZ00P2-180043. L. M. Polvani is funded by a grant from the NSF to Columbia University. The WACCM and SC-WACCM simulations were conducted on the National Center for Atmospheric Research
(NCAR) Cheyenne supercomputer.

## Appendix A

We calculate the NAM using a method similar to that of Gerber et al. (2010) and Gerber and Martineau (2018). The specific procedure is as follows:

1. We average model output to find a time series of daily, zonal mean geopotential height $Z(t, \lambda, p)$ as a function of time $t$, latitude $\lambda$, and pressure $p$.

2. For every day and pressure level, we remove the global mean geopotential height $\bar{Z}^{\mathrm{global}}(t, p)$. This helps to remove the global changes so that the index instead mainly captures meridional differences or shifts (Gerber et al., 2010). (While not the case for the simulations used in this study, this step would remove much of the global warming signal if it were present.)

3. For each day, latitude, and pressure level, we remove the average for that calendar day over the whole period; that is, we remove the climatology to find an anomalous height.

4. For each day, latitude, and pressure, we remove the linear trend over the period.

5. For each day and pressure level, we compute a polar cap average. Here we are interested in the NAM, and we take the average from 65-90°N. This is a proxy for the annular mode as shown in Baldwin and Thompson (2009).

6. We multiply by -1 so that a positive polar cap geopotential height anomaly yields a negative NAM, for consistency with the convention of Thompson and Wallace (1998).

7. We normalize the index by its standard deviation at each pressure level.

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

460

# U at 10 hPa

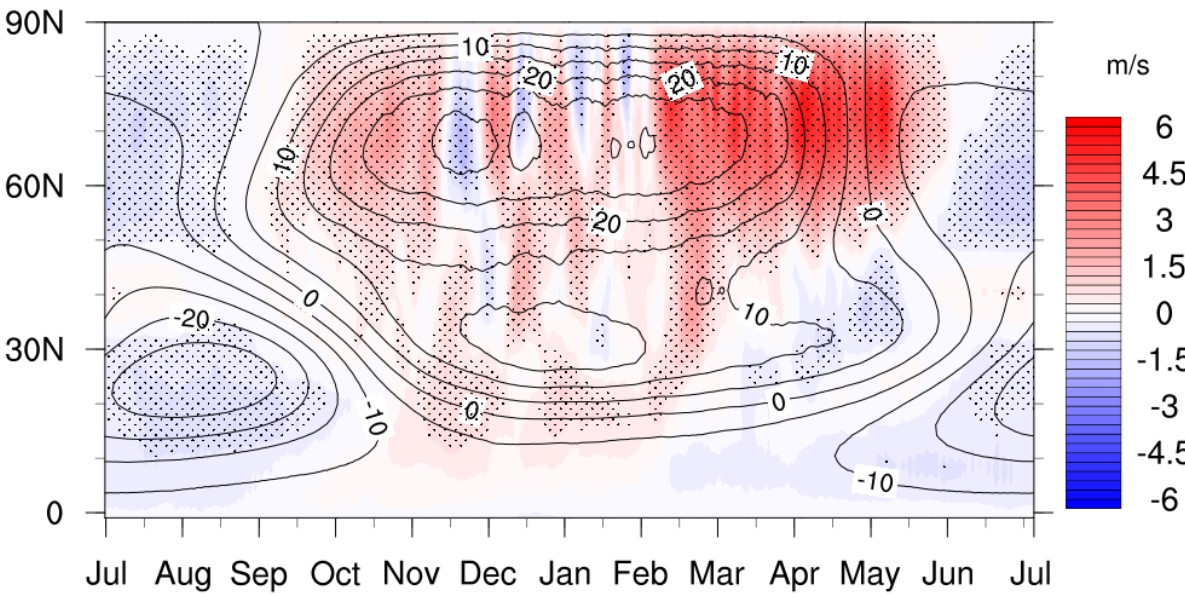

**Figure 1.** Latitude-time plot of zonal mean zonal wind at 10 hPa. Contours show NOCHEM values in m/s. Colored shading shows the difference CHEM-NOCHEM in m/s. Stippling indicates a significant CHEM-NOCHEM difference at a 95% level using a two-sample, two-tailed Welsh's t-test.

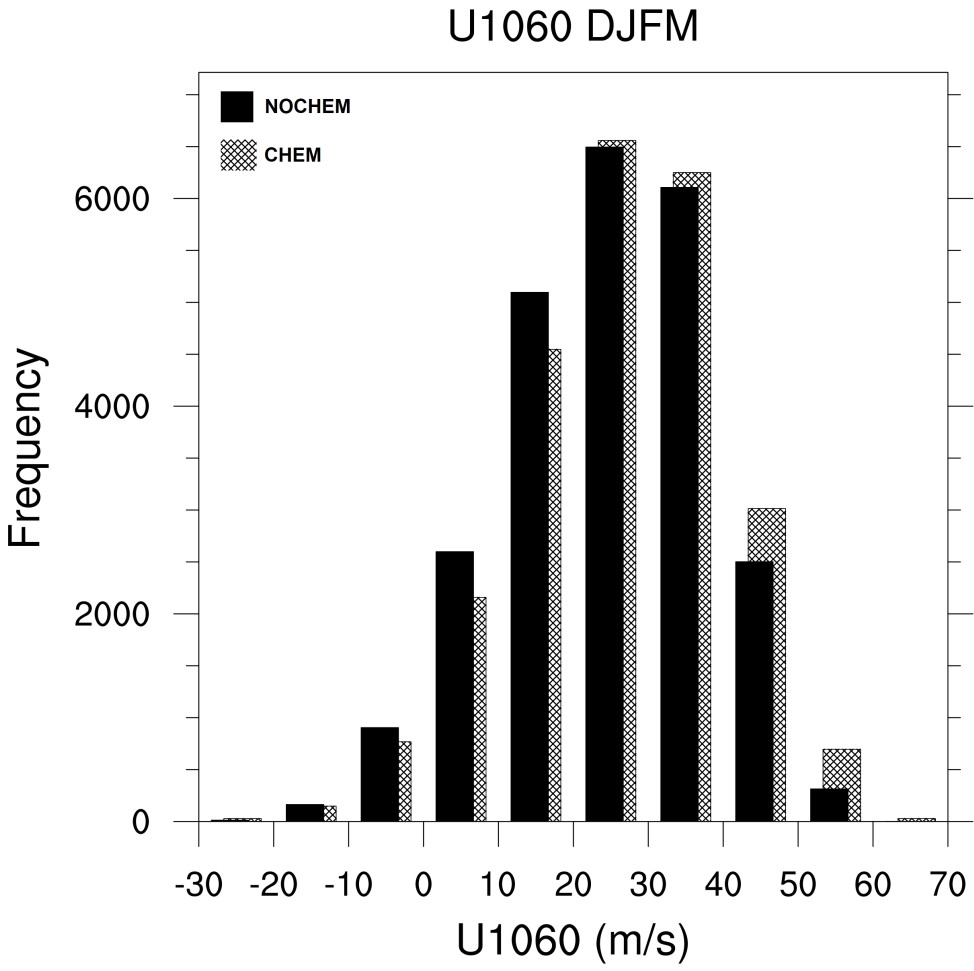

**Figure 2.** Histogram of daily values of zonal mean zonal wind at 10 hPa and 60°N in December-March for CHEM and NOCHEM. The mean of the CHEM and NOCHEM zonal mean zonal wind values are 26.1 m/s and 24.4 m/s respectively. The right tail of the CHEM distribution is longer, indicating more days of a particularly strong polar vortex.

|                  | NOCHEM | CHEM | Percent Difference | p-value |
|------------------|--------|------|--------------------|---------|
| Total Winters    | 200    | 200  |                    |         |
| DJF SSW events   | 75     | 67   | -10.7%             | 0.45    |
| DJF SPV events   | 58     | 74   | +29.3%             | 0.13    |
| March SSW events | 28     | 39   | +39.3%             | 0.14    |
| March SPV events | 7      | 5    | -28.6%             | 0.58    |

**Table 1.** Summary of sudden stratospheric warming (SSW) and strong polar vortex (SPV) events in 200-year year 2000 timeslices with and without interactive chemistry (CHEM and NOCHEM respectively). We separately consider the events occurring in December through February and those occurring in March. Reported p-values are based on a two-tailed two sample t-test (Charlton and Polvani, 2007b).

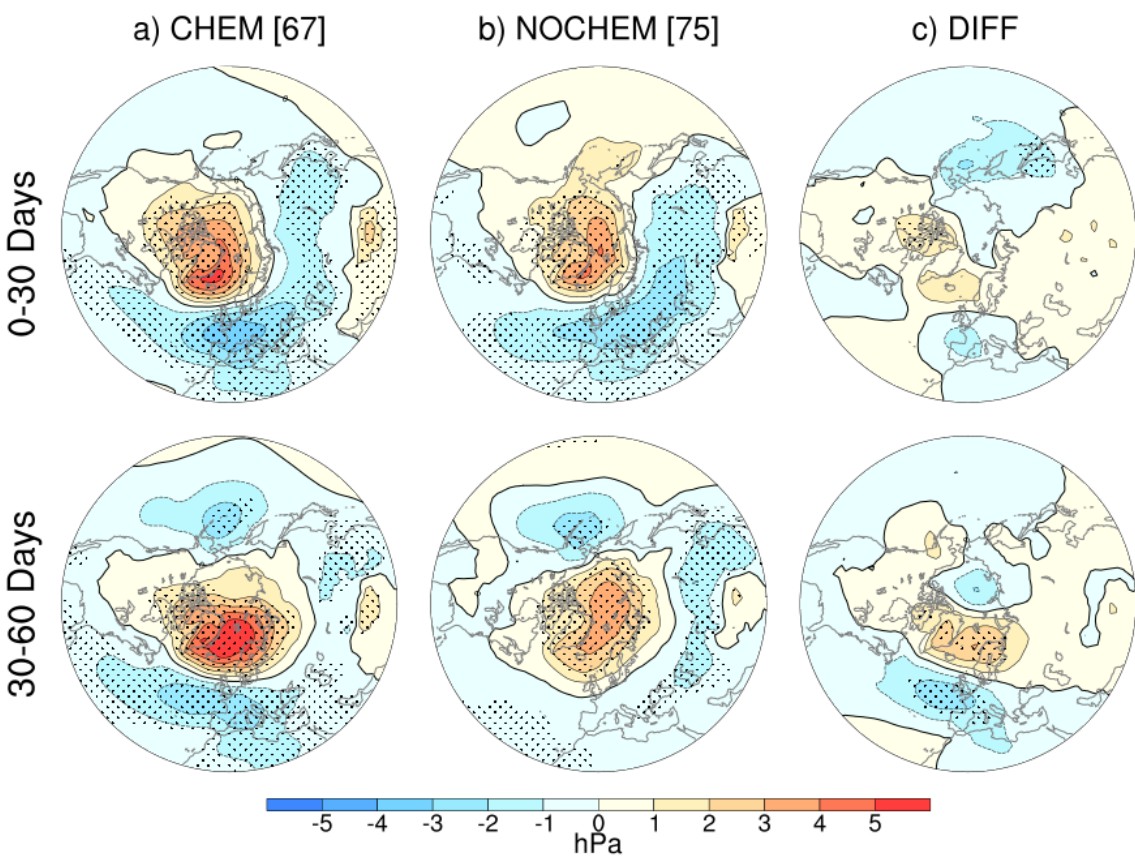

**Figure 3.** Composites of sea level pressure (SLP) anomalies (in hPa) in the 0-30 and 30-60 days following the central date of DJF SSWs in CHEM (WACCM, a) and NOCHEM (SC-WACCM, b) simulations, as well as the difference in the CHEM and NOCHEM composites (c). Significance at the 95% level using a Monte Carlo test (a,b) or a two-sided t-test (c) is indicated by stippling. The number of events included in each composite is noted in brackets above the figures.

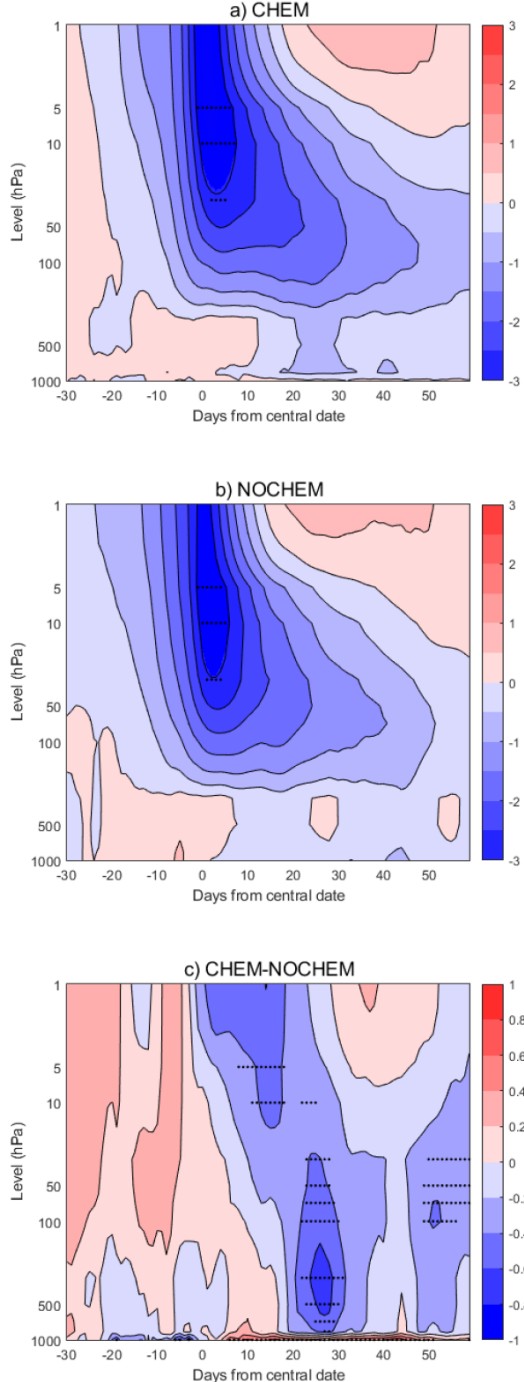

**Figure 4.** NAM anomaly composites around DJF SSW central dates in CHEM (a), NOCHEM (b), CHEM-NOCHEM (c). Stippling shows significance at the 95% level (with a Monte Carlo test for CHEM and NOCHEM and a two-tailed t-test for CHEM-NOCHEM). Contours are every 0.5 standard units for CHEM and NOCHEM and every 0.2 standard units for CHEM-NOCHEM.

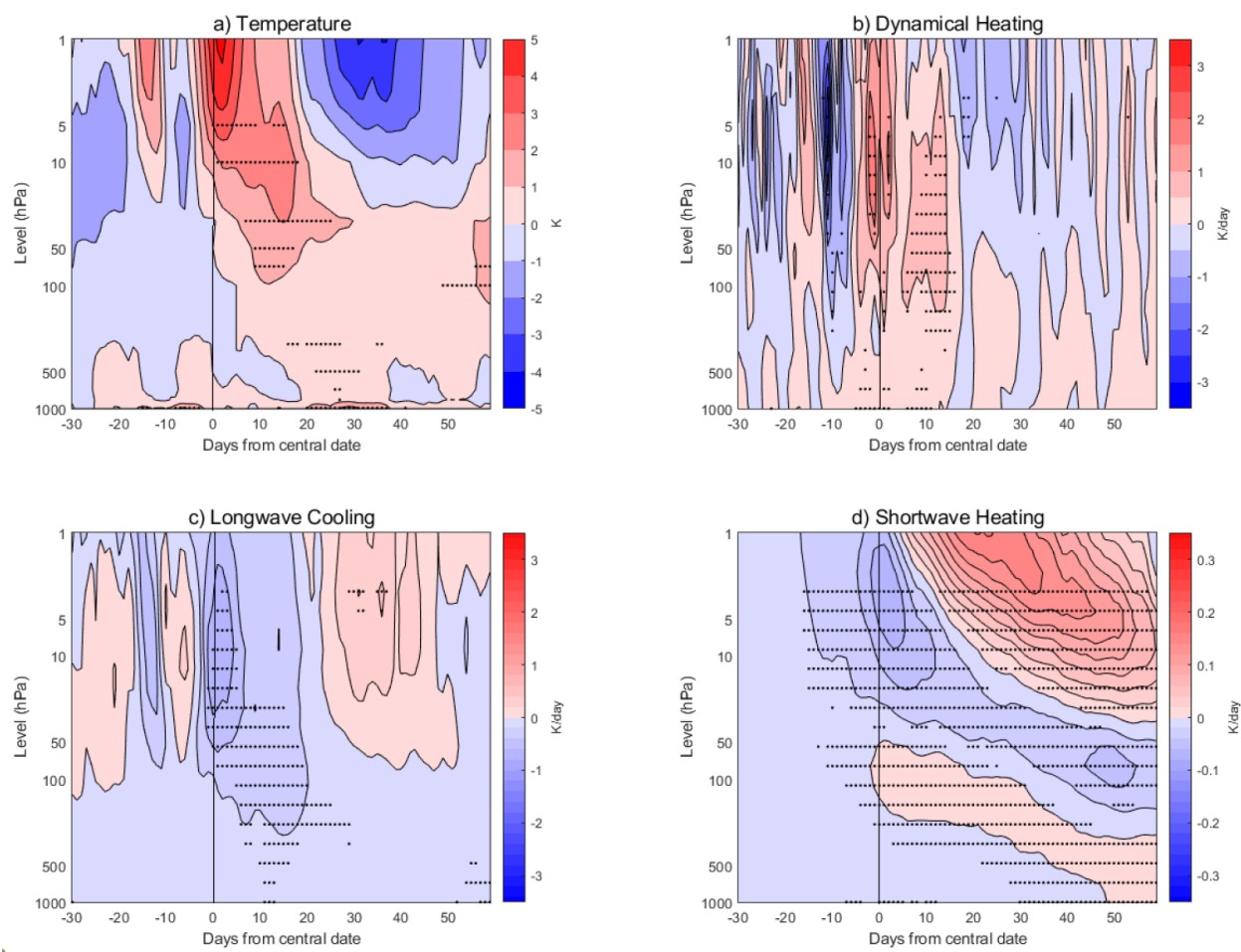

**Figure 5.** CHEM-NOCHEM differences in the temperature and heating anomalies over 60-90° N from -30 to +60 days around the SSW DJF central dates. (a): Temperature anomalies. Contours are every 1 K. (b): Dynamical heating anomalies. Contours are every 0.5 K/day. (c): Longwave heating anomalies. Contours are every 0.25 K/day. (d): Shortwave heating anomalies. Contours are every 0.02 K/day. Stippling shows significance at the 95% level under a two-tailed t-test.

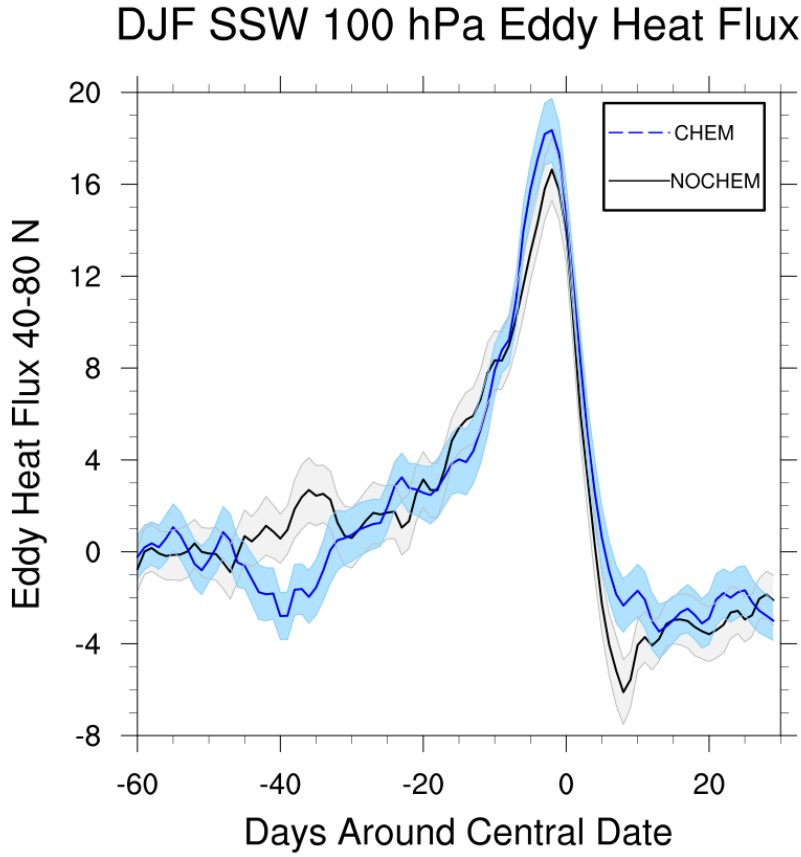

**Figure 6.** Eddy heat flux in mK/s over 40-80° N from -60 to +30 days around the SSW DJF central dates. The CHEM average is in blue, with confidence intervals shown in pale blue. The NOCHEM average is in black, with confidence intervals shown in gray.

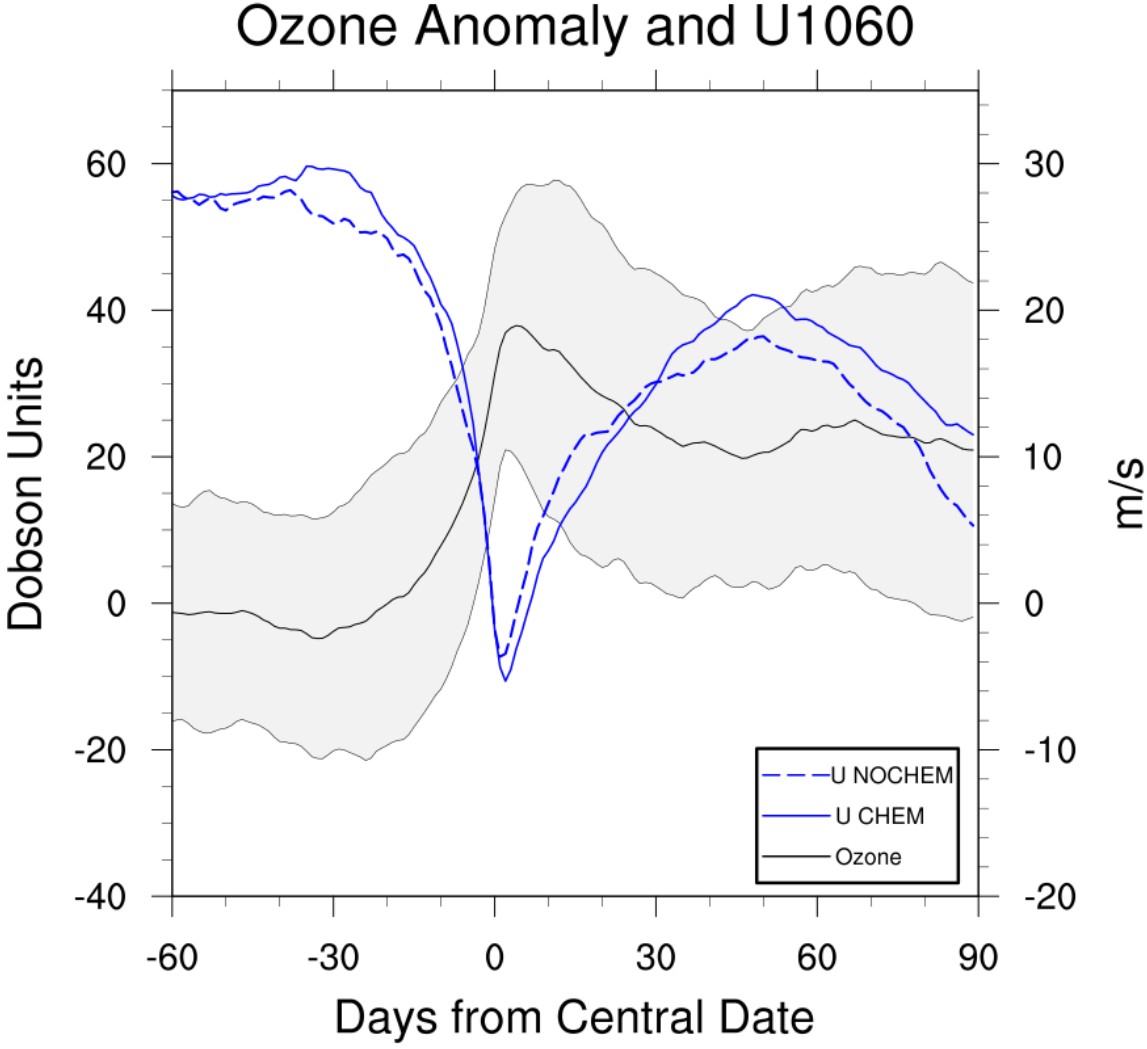

**Figure 7.** Composite of total column polar cap (over 60-90° N) ozone anomalies in Dobson units in the CHEM simulations and composites of zonal mean zonal wind at 60° N and 10 hPa in m/s from -60 to 90 days around the central date of DJF SSWs in CHEM and NOCHEM. The black line shows the mean total ozone column; $1\sigma$ from the mean is shaded. The blue solid and dashed lines shows the mean U1060 in CHEM and NOCHEM respectively.

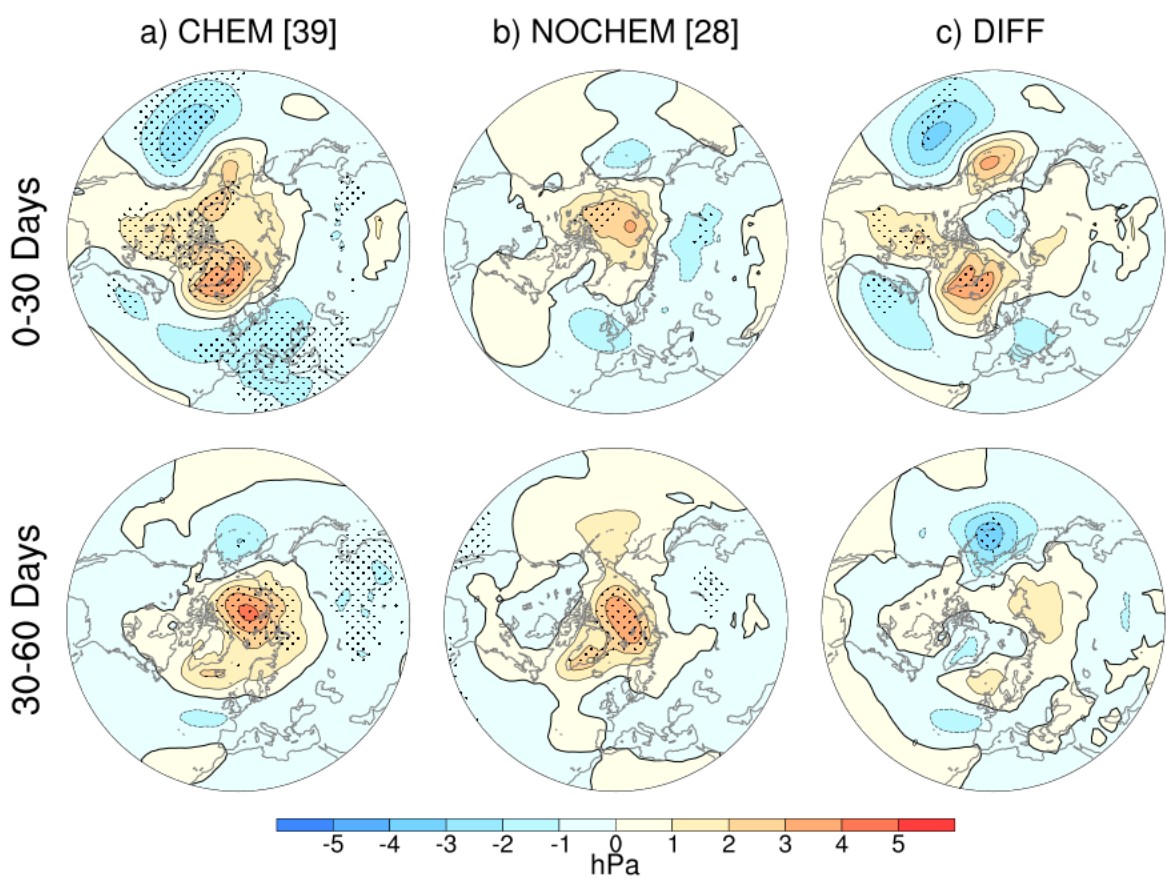

**Figure 8.** As in Figure 3, for March SSWs.

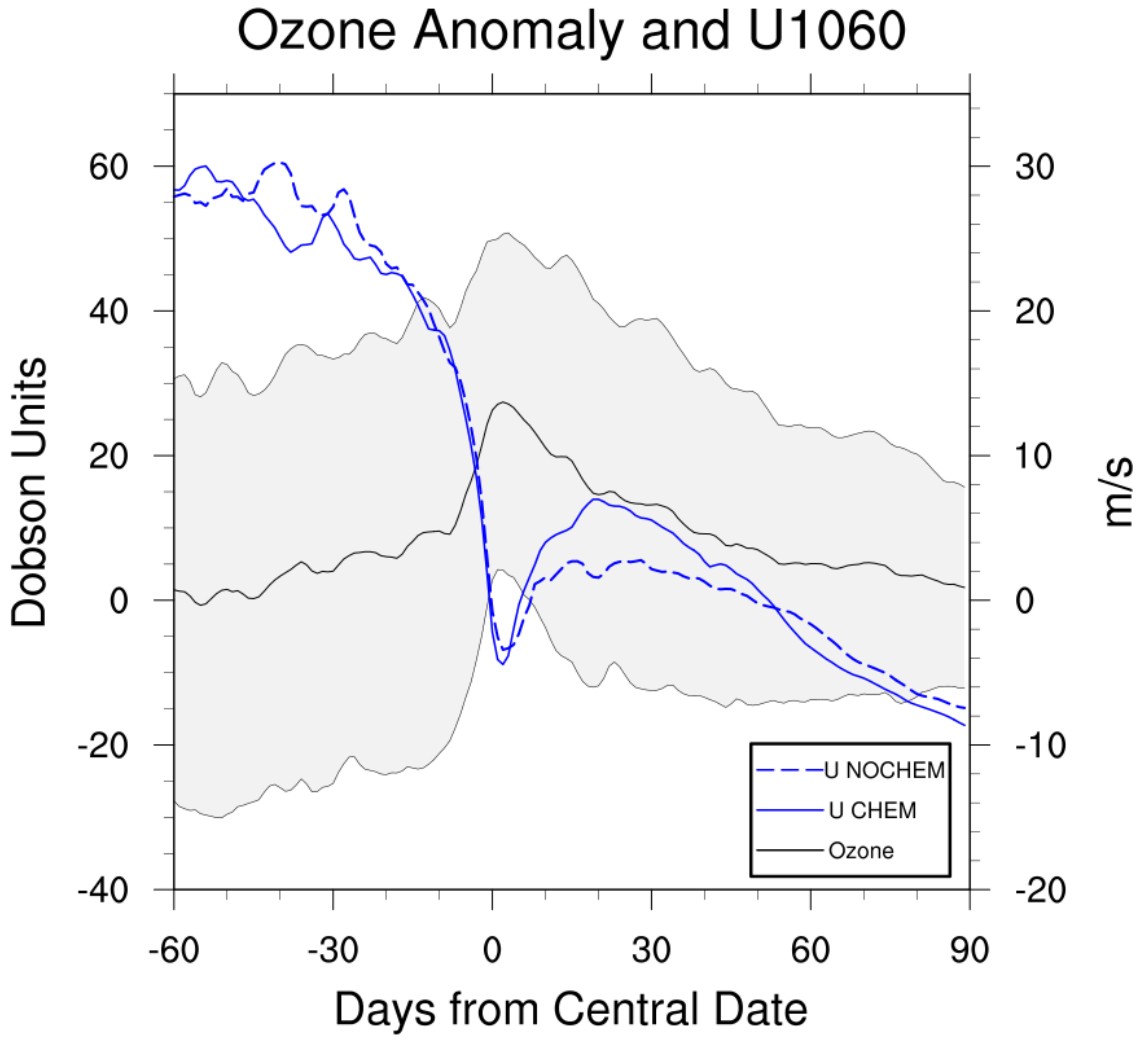

**Figure 9.** As in Figure 7, for March SSWs.

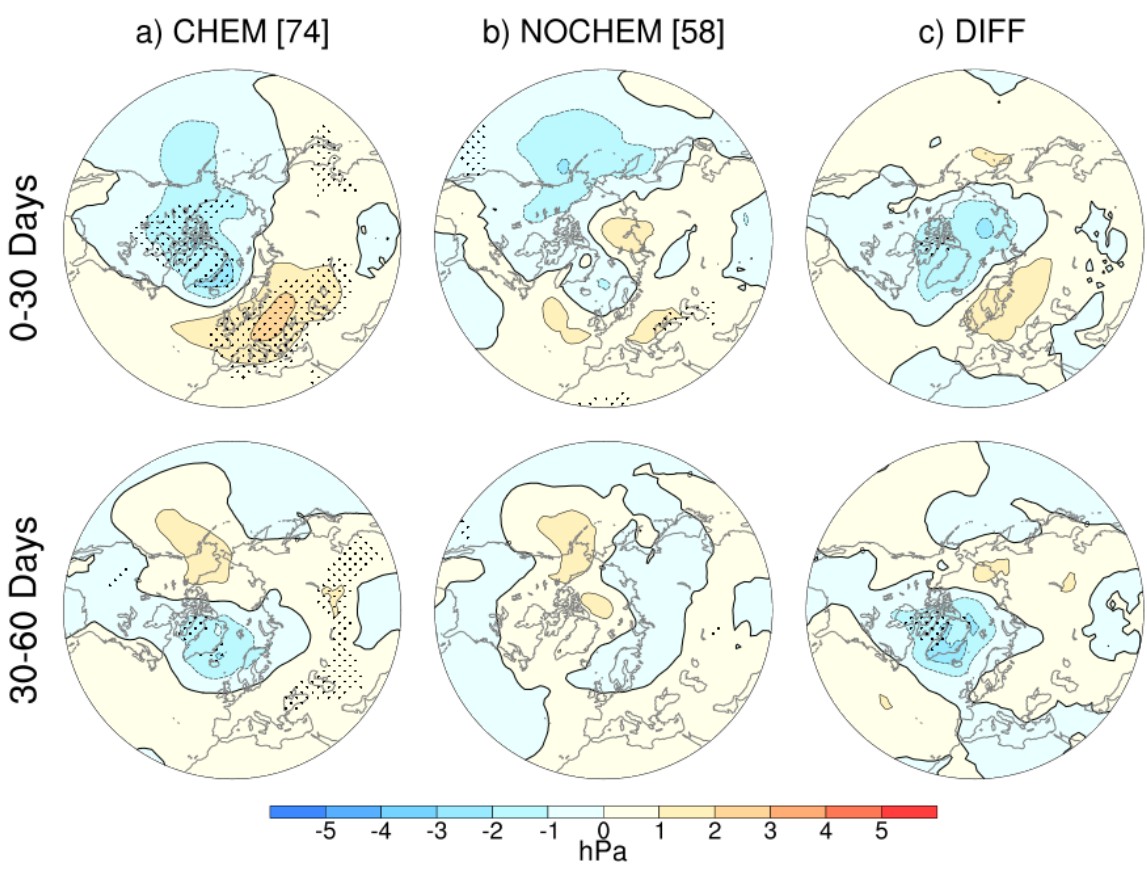

**Figure 10.** As in Figure 3, for DJF SPVs.

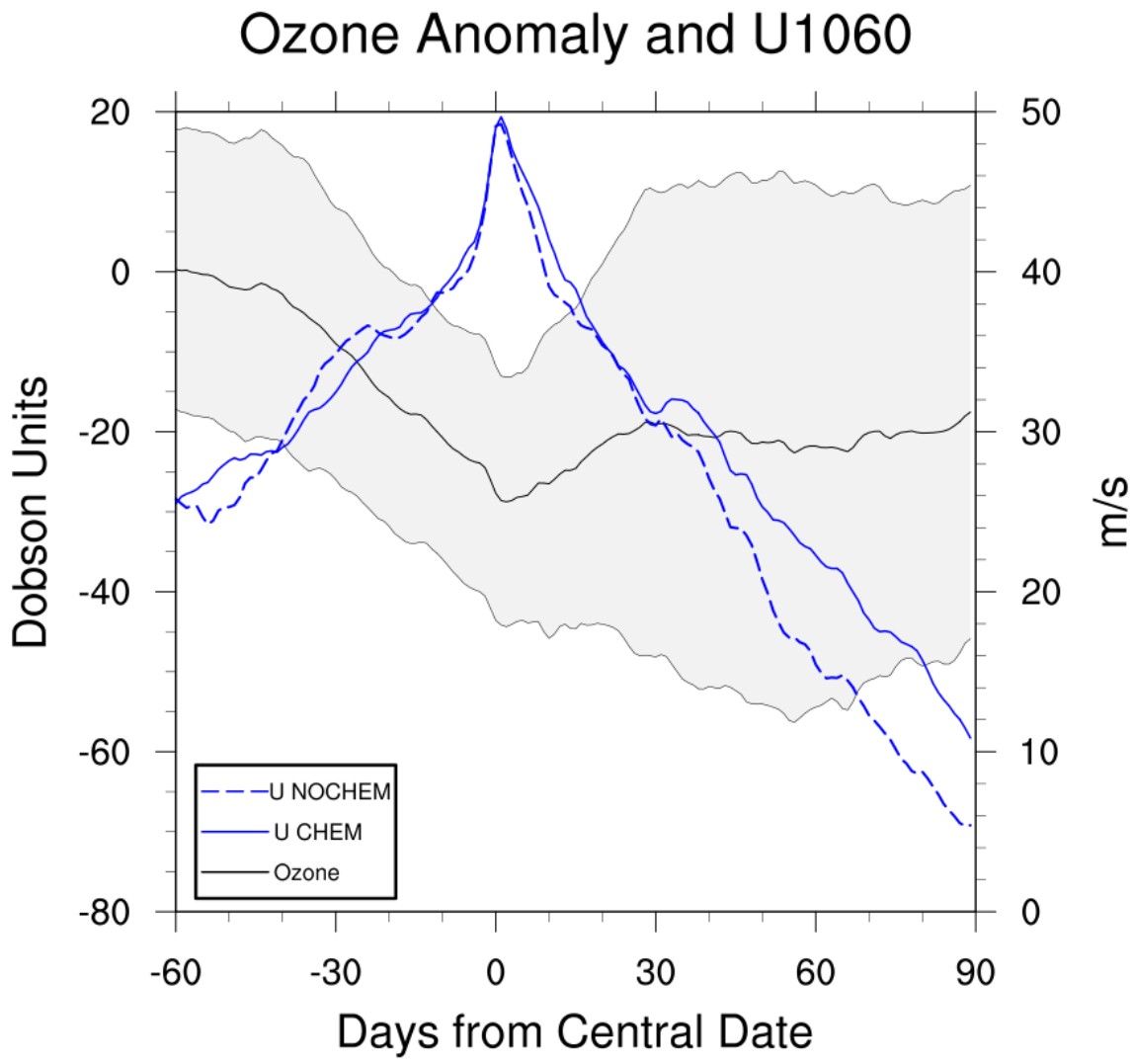

**Figure 11.** As in Figure 7, for DJF SPVs.