# Peer review of "The effect of interactive ozone chemistry on weak and strong stratospheric polar vortex events"

_Atmospheric Chemistry and Physics, 2020_

## Referee Comment (RC1) · Anonymous Referee #1 · 6 Apr 2020

General comment:

The paper investigates the effect of interactive ozone chemistry on the state of the polar vortex and downward coupling to the surface during Sudden Stratospheric Warmings (SSWs) and Strong Polar Vortex events (SPVs). The study is based on WACCM simulations using either interactive or specified chemistry. In particular, the analysis differentiates between mid winter (December-February) and late winter (March) events. It is found that in simulations with interactive chemistry the polar vortex is stronger. For mid winter warmings, a stronger and more persistent downward coupling occurs, characterized by more persistent NAO anomalies, in agreement with previous studies. For March warmings, and as well for SPVs, no significant changes occur. I regard this an interesting study, emphasizing the impact of interactive chemistry for simulations

of coupling between stratosphere and troposphere. The paper is well written and the presentation quality is good. I only have a few minor and technical remarks and do recommend the paper for publication.

Minor comments:

1. There is some inconsistency in the terminology to describe the two simulations. In Sect. 2 it is defined that WACCM=CHEM and SC-WACCM=NOCHEM. However, in some figures (e.g., Fig. 2, 3) again the terms WACCM and SC-WACCM are used, or even together with CHEM/NOCHEM. This is not a big issue, but I would recommend to stay with one clear terminology throughout the paper.

2. The referencing to the sub-figures (in the caption, and also in main text) should be made with a/b/c labels, not top/middle/bottom (e.g. for Figs. 4, 5, 6).

3. Overall, the paper has a lot of figures (15) with many of them showing insignificant differences. I think the presentation could become even clearer and more focussed on the main messages if the number of figures was somewhat reduced. E.g., Fig. 14 shows mainly insignificant differences (for case of SPVs), and in my opinion could be removed and related results just mentioned in the text. But I would leave it open to the authors to address this comment or not.

Technical comments:

P1, L20: ... tend to correspond to a ...

P2, L47: ... as their surface effects ...

P4, L116: ... 60°N ... The N is in italics.

P6, L170: ... temperatures ...

Figure 2: This is a case where there is some inconsistency in the terminology: The figure legend says SC-WACCM, WACCM, the caption says CHEM, NOCHEM.

Figure 4: This is a case where the referencing in the caption to the subfigures is not made with a/b/c labels.

[Figure]

---

## Referee Comment (RC2) · Anonymous Referee #2 · 8 Apr 2020

General Comments

This paper investigates differences in mid-winter sudden stratospheric warmings (SSWs), strong polar vortex events (SPVs), and March SSWs for timeslice simulations run with and without interactive chemistry. They find significant differences when interactive chemistry is included for mid-winter SSWs. Overall the study is clearly written and within the scope of ACP. I think the mechanisms for the differences they are seeing could be better explained, but in general I recommend the paper for publication if my comments can be addressed.

Specific Comments

Line 16: I'd be careful about the word "induce" here (maybe instead: "associated with").

For example, Ivy et al. 2017 states that causality is difficult to determine, as ozone could just be a proxy for dynamical effects.

Line 18: greater interannual variability of what? Also, in general it's a little tricky in these first few sentences of the introduction to understand if the focus is on the role of man-made ozone depletion or on the role of ozone extremes in general (regardless of the presence of CFCs).

Line 21-22: Particularly for the NH, where it's rarely cold enough for PSCs to form, what is the relative role of heterogeneous chemistry versus decreased mixing with mid-latitude air? I would expect mixing (or lack of it) to play a significant role.

Line 62: I'm not sure what is meant by "via its interannual variation".

Line 80: It would be good to be more explicit about what else is prescribed; given that you are using year-2000 conditions, for example, how are CFCs dealt with? (are they the same fixed value in each simulation? Also, do you think you need to be in a time period of peak CFCs to see the results that you get here? If true, that seems like an important point to make, especially as ozone is now slowly recovering.

Line 81: Could some of the differences seen in the results be due to using zonally-symmetric prescribed ozone, since during SSWs in particular the flow is very asymmetric? Could this also help explain why there is a larger difference in interactive vs prescribed for SSWs compared to SPVs (where the flow is very zonal)?

Line 92-93: Might also mention that surface impacts in March could be different than in mid-winter, as the NAO itself is changing in spatial structure from its winter to summer state.

Line 99: Is there any sensitivity to your results if you use a different threshold, such as 43 m/s? If you use a non-absolute metric like NAM, can you better examine differences in Dec-Feb versus March SPVs? It would be nice to be able to compare to Ivy et al. results, for example.

Line 105, and throughout: when you use the two-sided two-sample t-test, what are the sample sizes used (is it just the number of events)? If it's the latter, this seems valid for, e.g., Figure 3. But I wonder about Figure 1 in particular where you are showing daily data; in this case, what is the sample size for CHEM vs NOCHEM and does it take into account auto-correlation of the daily time series?

Line 113-115: So, what is the mechanism for the difference using year 2000 or year 1850 conditions? Is it just that you need CFC-driven ozone depletion to affect the basic state of the polar vortex?

Line 119: this difference seems extremely small given the large internal variability of the polar stratosphere zonal winds so I'm surprised that it's significant; how is significance assessed? Does it take into account auto-correlation of daily time series?

Line 123: Given that the climatology is different in these two simulations, I wonder if it's worth also checking the statistics of a non-absolute metric like the NAM, to see if the variance/PDF of the NAM changes.

Line 125: You might mention in this paragraph how well the overall (Nov-Mar) SSW frequency compares with reanalysis, as the Dec-Feb value seems low (75/200 = 0.38).

Line 132-133: I wonder also if this is because the CP07 definition is somewhat problematic with regards to the arbitrary April 30 cutoff for SSWs. Events in March essentially only have to pass a 10-day return to westerlies criteria (before April 30) to be counted as an SSW; when stricter criterion is used to separate these events from the final warming, some of them drop out (Butler and Gerber 2018).

Line 170, 187: Is it possible to demonstrate that there is an increase in dynamical forcing? This is mentioned several times as the mechanism but it's not shown that this is true.

Line 175-180: Can you explain further the vertical/temporal structure in the shortwave heating differences (Figure 5 and Figure 6)? Is it related to the vertical structure of

ozone after the SSW?

Line 198-199: so is it weaker SSWs (which doesn't seem true, see next comment below) than in DJF or weaker stratosphere-troposphere coupling or both? This seems easy enough to check. Another possibility is that the surface response in spring is just not as strong/significant as the NAO evolves to a summer state.

Line 215: True, but it's worth noting that the magnitude of the dynamical heating for March SSWs and DJF SSWs is about the same (at least on visual comparison). Does this imply that the March SSWs are not dynamically weaker than the DJF SSWs? (this seems to be stated on line 256). The minimum zonal wind reversal in Figure 7 and Figure 11 looks roughly equivalent, but the DJF SSWs start from a stronger westerly climatology. Again, it might be nice to show a dynamical metric here as well.

Line 246-247: I'm not sure it was ever explained clearly why the interactive chemistry simulation has a stronger basic state of the polar vortex.

Appendix A: just wanted to comment that this is a useful recipe.

Technical Corrections

Line 36: change "stratosphere" to "polar stratosphere"

Line 206: I'd remove "As for the surface plots,"

Line 242: change to "maximum absolute anomaly"

Line 263: change to "are important for accurately capturing the response"

Figure 1: can you make tick marks match up with the start of months, as it's difficult to infer where, for example, January starts.

Figure 2, 3, 8: Use CHEM and NOCHEM for consistency with text, rather than WACCM and SC-WACCM

---

## Referee Comment (RC3) · Anonymous Referee #3 · 11 Apr 2020

The goal of the study is to evaluate the effects of ozone chemistry on the stratosphere-troposphere coupling during SSWs and strong vortex events. To this aim, two different configurations of the chemistry climate model WACCM are used: CHEM, the standard version, and NOCHEM, a version with prescribed zonal mean profiles of ozone and other relevant species. Very much like Haase and Matthes (2019), the authors find differences in the basic state and the signal of mid-winter SSWs on surface climate variability. No relevant results are found for March SSWs or for strong vortex events.

The paper is well written and structured, and easy to follow. However, I find the new findings incremental. The results are basically reproducing those from Haase and Matthes (2019), using the same models but with longer runs and different external forcing. The paper also lacks a clear explanation of the mechanisms behind the differences in basic state and variability between CHEM and NOCHEM, they are simply attributed to "interactive chemistry". This could be true, but note that this is not the result of a forced experiment versus a control. This is essentially a comparison between two different models, one that has been exhaustively tuned at NCAR to provide the best possible climatology and variability (CHEM), and one that has been "downgraded" by specifying the evolution of ozone and other species (NOCHEM). The reported differences might be at least in part a consequence of this.

I think there is potential in the manuscript, but I cannot recommend publication in its present form. I will reconsider after major revision.

Specific comments:

- Line 18. Interannual variability of what?

- First paragraph of section 3.1. It should be good to at least speculate about a mechanism to explain the stronger westerlies in CHEM. In the light of the given comparison against Haase and Matthes and Smith et al, are high CFC concentrations needed to get this result?

- Line 119. I find it surprising that a 1.7 m/s difference be statistically significant, given the large interannual variability in the strength of the westerlies. But letting aside the statistics, how physically meaningful might this difference be?

- In line 153 it is stated that Fig. 3 reveals the surface impacts of SSWs. In the beginning of the following paragrah it is addressed whether the results of Fig.3 are due to SSWs. Please consider presenting first Fig. 4, and then Fig. 3 (or revise the way the story line is presented).

- Line 163. I am not so sure about this. The positive T signal in Fig. 5a appears at positive lags, generally after the peak of the warming. Without further analysis, this might well be interpreted as a slower temperature recovery in the aftermath of SSWs in CHEM than in NOCHEM.

- Line 168-169. "(...) perhaps because of greater wave activity necessary for an SSW to occur with a stronger mean vortex state (...)". The authors provide no proof of this, but it should be easy to check. Please consider plotting composite differences of U60 as a function of lag and pressure (to see if the vortex is stronger before SSWs in CHEM than in NOCHEM), and EP flux divergence over 45N-75N (to see whether there is stronger wave forcing in CHEM than in NOCHEM).

- In this same paragraph, an alternative explanation of stronger dynamical heating in CHEM could be that having a stronger jet in the mid-stratosphere generally implies a stronger vertical gradient of temperature over the pole (see Fig. 3b In Haase and Matthes). And a given w* anomaly, under a stronger temperature vertical gradient, would produce a stronger dynamical heating.

- Figure 5. It would be good to include the temperature tendency to better understand the effects and timing of the dynamical heating (please be explicit, is this w*dt/dz, TEM formalism?), and the radiative heating. Also, please include in the caption the latitude band considered.

- Line 170-172. So would it be possible to discriminate the fraction of the stronger long-wave cooling that comes from having reached higher temperatures, from the fraction that comes from having larger ozone concentrations? This would tell us the importance of the accumulation of ozone over the pole during SSWs to the recovery of the vortex.

- Figure 6. Please consider removing this figure, it does not really add much to the manuscript.

- Lines 183-190. The study of De La Camara et al also shows results on changes of polar ozone during SSWs using a similar version of WACCM as the one used here.

- Line 200. Please take into account that NAM decorrelation timescales are shorter in March-April than in winter (Simpson et al. 2011)âĄă, which may help understand the weaker persistence of surface anomalies after March SSWs.

- I am unsure whether it is worth keeping section 3.4, but 4 figures to only show a very weak signal seems excessive. - Conclusions, first sentence. This has not been shown, at least to my understanding. A mechanism behind the circulation differences between CHEM and NOCHEM has not been provided, nor have the authors discriminated between the effects of ozone chemistry and the effects of transport of ozone during SSWs.

- Line 253-254. It is a speculation that CHEM needs stronger wave forcing to drive SSWs, it has not been proven. Please modify this sentence accordingly.

Technical comments:

- Please refer to individual figure panels using the labels a, b, c, etc. and not top-right etc.

References:

Simpson, I. R., P. Hitchcock, T. G. Shepherd, and J. F. Scinocca, 2011: Stratospheric variability and tropospheric annular-mode timescales. Geophys. Res. Lett., 38, L20806, doi:10.1029/2011GL049304. http://doi.wiley.com/10.1029/2011GL049304.

---

## Author Comment (AC1) · 19 Jun 2020

We thank the reviewers for their helpful comments. We have addressed the issues raised by the reviewers to the best of our abilities. In particular, we have included more dynamical metrics as suggested by Referees 2 and 3 to strengthen our discussion of the potential mechanisms of the results studied here. We have corrected or clarified some statistical testing, also suggested by Referees 2 and 3. We have moved several figures to supplementary material at the suggestion of Referees 1 and 3 and have pared down discussion of these figures in the text. This has led to a more succinct presentation of our results.

We provide detailed replies to referees, with referee comments in bold and within angle

brackets and our responses and changes below. The marked-up manuscript version is included as a supplement.

Replies to Referee 1

<<

**There is some inconsistency in the terminology to describe the two simulations.**

>>

We have corrected Figures 2, 3, 8, and 10 to use CHEM and NOCHEM, as is used in the rest of the paper.

<<

**The referencing to the sub-figures (in the caption, and also in main text) should be made with a/b/c labels, not top/middle/bottom.**

>>

We have corrected this throughout the text and in figure captions.

<<

**Overall, the paper has a lot of figures (15) with many of them showing insignificant differences. I think the presentation could become even clearer and more focussed on the main messages if the number of figures was somewhat reduced.**

>>

We thank the reviewer for this recommendation. We have moved 5 of the original figures to supplementary material (now Supplementary Figures 1, 2, 4, 5, and 6), mainly

those showing insignificant differences, and have tightened discussion of these figures.

The reviewer also included several technical comments; we have corrected typos or wording accordingly.

Replies to Referee 2

<<

**Line 16: I'd be careful about the word "induce" here (maybe instead: "associated with"). For example, Ivy et al. 2017 states that causality is difficult to determine, as ozone could just be a proxy for dynamical effects.**

>>

We agree and have followed the referee's recommended wording. See Lines 15-16.

Line 15-16: "Ozone extremes have also been shown to be associated with springtime surface anomalies in the Northern Hemisphere."

<<

**Line 18: greater interannual variability of what? Also, in general it's a little tricky in these first few sentences of the introduction to understand if the focus is on the role of man-made ozone depletion or on the role of ozone extremes in general (regardless of the presence of CFCs).**

**Line 21-22: Particularly for the NH, where it's rarely cold enough for PSCs to form, what is the relative role of heterogeneous chemistry versus decreased mixing with mid-latitude air? I would expect mixing (or lack of it) to play a significant role.**

>>

We have clarified the use of "interannual variability." We intended the focus of this introduction to be on the role of ozone extremes in general, and have rewritten this paragraph to make this clearer, though some mechanisms specific to the presence of CFCs are still present (i.e. episodic ozone depletion events such as 2011). We agree with the referee about the role of mixing and have included discussion of this mechanism. See updated discussion on Lines 18-24.

Lines 18-24: "Polar cap ozone anomalies are strongly related to interannual variability in stratospheric polar vortex strength, which is larger in the Northern Hemisphere than the Southern Hemisphere. This is a result of the larger amplitudes of upward-propagating planetary waves, which perturb the stratospheric circulation. Years with low wave activity tend to correspond to a stronger vortex and a weaker Brewer-Dobson Circulation (BDC), resulting in weaker ozone transport from the tropics into the poles and decreased mixing across the vortex edge, as well as enhanced formation of polar stratospheric clouds, which contribute to increased springtime destruction of ozone. Years with high wave activity correspond to a weaker vortex and a stronger BDC, with stronger ozone transport from the tropics and increased mixing (Newman et al., 2001)."

<<

**Line 62: I'm not sure what is meant by "via its interannual variation".**

>>

We have rephrased this sentence. See Lines 62-63.

Lines 62-63: "Due to the larger sample size, climatological ozone distribution, and constant forcings, this set of simulations more clearly separates the impact of ozone's interannual variation on stratosphere-troposphere coupling."

<<

**Line 80: It would be good to be more explicit about what else is prescribed; given that you are using year-2000 conditions, for example, how are CFCs dealt with? (are they the same fixed value in each simulation? Also, do you think you need to be in a time period of peak CFCs to see the results that you get here? If true, that seems like an important point to make, especially as ozone is now slowly recovering.**

>>

Alongside ozone , CFCs (CFC-11 and CFC-12), methane ($CH_4$), nitrous oxide ($N_2O$), and $CO_2$ are prescribed at year 2000 conditions in the NOCHEM (SC-WACCM) simulation. Other fields that are prescribed in this run are NO, atomic and molecular oxygen (O and $O_2$), following Smith et al. (2014). In the CHEM experiment, we use the same year 2000 conditions for all radiatively active gases, but ozone is calculated interactively, instead of being specified. We clarify this point on Line 82 and Lines 84-85. We speculate about the role of CFCs in Lines 122-127.

Lines 81-85: "In the NOCHEM simulation, ozone concentrations (and other radiatively active atmospheric constituents, including CFCs) are prescribed using zonally symmetric, monthly mean, seasonal climatology computed from the WACCM integration. These zonally symmetric monthly ozone fields are read into SC-WACCM and interpolated linearly to the day of the year. More details can be found in Smith et al. (2014). Hence, both CHEM and NOCHEM strictly impose identical year 2000 forcings for all radiatively active species, and only differ in their treatment of ozone."

Lines 122-127: "The same is not the case in Smith et al. (2014), where the vortex under constant year 1850-conditions was found to be of similar strength with interactive and specified chemistry. The difference in strength of the vortex with interactive vs. specified chemistry is then partially related to forcings used (year 2000 in this study, 1955-present historical in Haase and Matthes (2019), and 1955-2005 historical

in Neely et al. (2014)); this suggests a potential relationship to CFCs. The details of this will be investigated in a further study, but this may be related to lower ozone variability in the absence of CFCs (Calvo et al., 2015)."

$<<$

**Line 81: Could some of the differences seen in the results be due to using zonally-symmetric prescribed ozone, since during SSWs in particular the flow is very asymmetric? Could this also help explain why there is a larger difference in interactive vs prescribed for SSWs compared to SPVs (where the flow is very zonal)?**

$>>$

Yes, this is very possible. There is some indication that the differences in midwinter SSWs are partially related to zonally-symmetric ozone; Haase and Matthes (2019) found lower differences when prescribing ozone asymmetrically, though it did not erase all interactive vs. prescribed differences. We have added discussion of this and why we still consider zonally-symmetric prescribed ozone in Lines 86-91.

Lines 86-91: "One might consiser specifiying non-zonally symmetric ozone (Haase and Matthes, 2019), but that comes at cost of a major physical incosistency between the polar vortex and the ozone field: in other words the exteme ozone years in the model will not correspond with the unperturbed vortex years. More importantly, the vast majority of climate models in CMIP specificy zonally symmetric stratospheric ozone, including within CMIP6 (Keeble et al., 2019): hence the zonally-symmetric specified ozone case is the one of most interest in terms of evaluating the impact of interactive ozone chemistry."

$<<$

**Line 92-93: Might also mention that surface impacts in March could be different than in mid-winter, as the NAO itself is changing in spatial structure from its winter to summerstate.**

$>>$

Yes, this is quite plausible. We have added a mention of this on Lines 99-100.

Lines 99-100: "We consider March events separately from December-February events due to different shortwave heating behavior, model bias in March SSW frequency (too frequent SSWs in our model), and different NAO structure in early spring compared to winter."

$<<$

**Line 99: Is there any sensitivity to your results if you use a different threshold, such as 43 m/s? If you use a non-absolute metric like NAM, can you better examine differences in Dec-Feb versus March SPVs? It would be nice to be able to compare to Ivy et al. results, for example.**

$>>$

We redid key parts of the analysis using the 41.2 m/s threshold from Tripathi et al. 2015. This adds 10 more SPV events in NOCHEM and 11 more events in CHEM, so there is little change to the frequency results. There was also little change to the SLP results. We agree that a non-absolute metric would allow for better study of strong vortex events in March; for consistency with the rest of this work and the most standard SSW metrics, in the paper we only consider an absolute metric. See clarification on Line 108.

Line 108: "Results are not sensitive to using a 41.2 m/s threshold as in Tripathi et al. (2015)."

<<

**Line 105, and throughout: when you use the two-sided two-sample t-test, what are the sample sizes used (is it just the number of events)? If it's the latter, this seems valid for, e.g., Figure 3. But I wonder about Figure 1 in particular where you are showing daily data; in this case, what is the sample size for CHEM vs NOCHEM and does it take into account auto-correlation of the daily time series?**

>>

The sample sizes used are indeed the number of events. In later figures (Figure 3 and following), this is the number of the type of SSW/SPV under consideration. In Figure 1, it is the number of years in the simulation. At each time/latitude point in Figure 1, we consider 200 samples from each of CHEM and NOCHEM, treating each year of the simulation as independent. While consecutive days in the simulation are auto-correlated, each t-test conducted here is on samples that we can treat as independent. The results of t-tests on consecutive days do have some correlation due to auto-correlation of the daily time series, but so do results of t-tests at adjacent latitude points due to spatial auto-correlation in the data. Because each individual t-test is on independent samples, and each t-test uses disjoint data, we do not correct for time or spatial auto-correlation. This may lead to over-confident significance test in Fig.1, but we believe that the core result of the paper (i.e. a generally stronger vortex in CHEM than NOCHEM) is not affected, as explained next.

<<

**Line 119: this difference seems extremely small given the large internal variability of the polar stratosphere zonal winds so I'm surprised that it's significant; how is significance assessed? Does it take into account auto-correlation of daily time series?**

>>

We thank the reviewer for their question about how significance was assessed in the polar stratospheric zonal winds. We were not properly accounting for auto-correlation of the daily time series here. While the figure still shows daily winds (to better show the extreme values), we test significance by considering average winds over each winter (DJFM) in the simulations. The average zonal mean zonal wind from one winter to the next is independent, so we can treat these winters as independent samples and use a Welsh's t-test. A two-tailed t-test gave a p-value of 0.023, so the difference is small but significant (due to the large sample size in each simulation). We have added a discussion of this issue on Lines 131-134.

Line 131-134: "To determine whether this is statistically significant, we consider the average zonal mean zonal winds over each winter and treat the winters as independent. A two-sample, two-tailed Welsh's t-test of DJFM average winds in CHEM and NOCHEM yields a p-value of 0.023, so the difference, though small, is significant at a 95% level."

$<<$

**Line 123: Given that the climatology is different in these two simulations, I wonder if it's worth also checking the statistics of a non-absolute metric like the NAM, to see if the variance/PDF of the NAM changes.**

$>>$

The variances of the daily 10 hPa NAM distributions in DJF are similar: 2.42 in NOCHEM and 2.32 in CHEM (with both averages 0 by design). There are some minor differences in the PDFs, but they are much more similar overall than are the zonal mean zonal wind distributions.

$<<$

**Line 125: You might mention in this paragraph how well the overall (Nov-Mar) SSW frequency compares with reanalysis, as the Dec-Feb value seems low (75/200 = 0.38).**

$>>$

We have added a comment based on this suggestion near Lines 150-152.

Lines 150-152: "We note that both of these frequencies, around 5.5 events per decade, are on the lower end of what is seen across reanalyses (Butler et al., 2017; Cao et al., 2019) but very well within the spread among state-of-the-art chemistry-climate models (Ayarzagüena et al., 2018)."

$<<$

**Line 132-133: I wonder also if this is because the CP07 definition is somewhat problematic with regards to the arbitrary April 30 cutoff for SSWs. Events in March essentially only have to pass a 10-day return to westerlies criteria (before April 30) to be counted as an SSW; when stricter criterion is used to separate these events from the final warming, some of them drop out (Butler and Gerber 2018).**

$>>$

Using the stricter criterion used in Butler and Gerber 2018, we find 20 fewer SSWs in CHEM; 19 of those removed are during March. We also find 20 fewer SSWs in NOCHEM; 15 of those removed are during March. This more stringent criterion would still give us 13 NOCHEM vs. 20 CHEM March SSWs, still a very notable difference. We hypothesize that this is due to the stronger vortex in the climatology and the resulting later breakdown of the vortex (giving more "room" for springtime SSWs to happen).

Under the weaker CP07 criterion used in the paper, we can see the vortex recover more strongly following a March SSW in CHEM than in NOCHEM in the wind time

series shown in Figure 9. The vortex isn't particularly strong in either case (and it's clear why the stricter criterion removes so many events!), but there's a meaningful difference in CHEM and NOCHEM in the aftermath of SSWs here.

$<<$

**Line 170, 187: Is it possible to demonstrate that there is an increase in dynamical forcing? This is mentioned several times as the mechanism but it's not shown that thisis true.**

$>>$

We have added a plot of the meridional eddy heat flux at 100 hPa, averaged between $40°$ and $80°$ N around the SSW central date as a new Figure 6, as well as a discussion of this figure. This is just a diagnostic, but it does indicate slightly stronger dynamical forcing in CHEM compared to NOCHEM, at least for midwinter SSWs. See discussion on Lines 189-194.

Lines 189-194: "The increased dynamical heating in CHEM could be related to greater wave activity necessary for an SSW to occur with a stronger mean vortex state. Figure **??** shows the eddy heat flux over 40-80° N over time in CHEM and NOCHEM. This is stronger by about 2 mK/s just before the central date in CHEM than in NOCHEM, indicating stronger wave forcing in CHEM. The CHEM and NOCHEM means are at the upper and lower bounds of the other's confidence intervals, respectively. Further, the zonal mean zonal winds at 10 hPa and 60° N around the central date of the SSW (shown in Figure **??**) are both stronger prior to the event and more easterly following the central date in CHEM than in NOCHEM."

$<<$

**Line 175-180: Can you explain further the vertical/temporal structure in the short-**

**wave heating differences (Figure 5 and Figure 6)? Is it related to the vertical structure of ozone after the SSW?**

$>>$

Yes, they are related. In fact, the SW differences are entirely originated by the CHEM run (interactive ozone), as the NOCHEM run shows no SW heating anomalies around SSWs, due to the prescribed ozone climatology. We have added a reference for the vertical structure of ozone after the SSW with a similar structure to the shortwave heating here on Lines 200-201.

Lines 200-201: "The structure in height and time is related to integrated effects of the ozone anomalies following the SSW, which show a similar structure (Kieswetter et al., 2010)."

$<<$

**Line 198-199: so is it weaker SSWs (which doesn't seem true, see next comment below) than in DJF or weaker stratosphere-troposphere coupling or both? This seems easy enough to check. Another possibility is that the surface response in spring is just not as strong/significant as the NAO evolves to a summer state.**

**Line 215: True, but it's worth noting that the magnitude of the dynamical heating for March SSWs and DJF SSWs is about the same (at least on visual comparison). Does this imply that the March SSWs are not dynamically weaker than the DJF SSWs? (this seems to be stated on line 256). The minimum zonal wind reversal in Figure 7 and Figure 11 looks roughly equivalent, but the DJF SSWs start from a stronger westerly climatology. Again, it might be nice to show a dynamical metric here as well.**

$>>$

The meridional eddy heat flux anomaly leading up to a March SSW in the CHEM simulation is lower than that seen leading up to DJF SSWs, which does suggest a dynamically weaker SSW in that case. The differences in NAM descent indicate a generally weaker stratosphere-troposphere coupling following March SSWs, as compared to DJF SSWs. So, both seem to be at play here. We have added discussion of different surface response due to seasonality as suggested, on Lines 218-220, and Lines 228-229.

Lines 218-220: "Three factors could contribute to this: weaker SSWs, weaker stratosphere-troposphere coupling, both discussed below, and a shorter NAM decorrelation timescale in March than in DJF (Baldwin et al., 2003; Simpson et al., 2011), which would result in weaker anomalies at the surface when averaged over several weeks."

Lines 228-229: "The eddy heat flux show in Figure S3, however, shows weaker wave forcing preceding only the CHEM (not the NOCHEM) March SSWs compared to those in DJF."

$<<$

**Line 246-247: I'm not sure it was ever explained clearly why the interactive chemistry simulation has a stronger basic state of the polar vortex.**

$>>$

This is correct. We have reworded based on this comment and a similar one from Referee 3 so that this sentence no longer implies a mechanism. See Lines 273-274.

Lines 273-274: "The climate model results presented here show an important relationship between interactive ozone, the climatological state of the stratospheric polar vortex, and the Euro-Atlantic surface impacts of midwinter SSWs."

We have also made phrasing and figure labeling changes based on the referee's technical comments. We have added clearer tick marks to Figure 1 and edited figure labels

throughout to be consistent with the text. The suggested phrasing changes have been incorporated throughout.

Replies to Referee 3

<<

**General comments: No relevant results are found for March SSWs or for strong vortex events.**

>>

We agree with the referee that there are no significant differences, but a lack of difference is a result. This null result is very much worth reporting here, given its implications. For example, we wish to discuss March SSWs (1) and the frequency of SSWs (2). The lack of significant differences in March SSWs is a relevant finding, as it challenges the expectation that ozone would have a larger signal in late-winter/spring (given the larger SW feedback in those months). The weak impact of ozone chemistry on the modeled SSW frequency is another key "no result", since it implies that interactive chemistry may not be needed to simulate a realistic SSW frequency in high-top models (possibly, the vertical resolution and model lid are more important parameters for this metric). We also challenge the referee's view concerning the strong vortex events: the vortex decay in the lower stratosphere 40-50 days after the onset of these events is slower in CHEM than NOCHEM, as discussed in Line 260-262; this means that extreme vortex events are longer-lived when ozone is interactive. Even though these effects are not directly relevant for surface climate, they have implications for our understanding of these dynamical events in the stratosphere.

<<

**I find the new findings incremental. The results are basically reproducing those**

**from Haase and Matthes (2019), using the same models but with longer runs and different external forcing.**

$>>$

We politely disagree with the referee concerning the degree to which these results are "incremental" over Haase and Matthes (2019; hereafter H&M19). First, we use time-slice experiments with constant forcings for the year 2000, whereas H&M19 use transient boundary conditions. Second, we have a 4x larger sample size in both CHEM and NOCHEM runs (200 years vs 50 years in H&M19). Third, we impose a climatological ozone distribution, whereas H&M19 impose a transient ozone. Fourth, we explore the effects in both polar vortex extremes, i.e. SSWs and strong vortices, whereas H&M19 only focus on the former. Finally, while we independently cofirm some of the results in Haase and Matthes (2019), we also report that some of their findings are not robust.

Owing to the larger sample size and cleaner model set-up (1+2), we are able to show that interactive ozone *does not affect* the modeled SSW frequency integrated over the extended winter (DJFM), which is in contrast with H&M19. We also think that our set-up (3) more cleanly isolates the effect of interactive ozone, as imposing a climatological ozone distribution in NOCHEM which does not contain any (dynamically induced) inter-annual variations is a more consistent way of removing the effects of ozone variability on the dynamical state of the stratosphere, as noted on Line 63. Lastly, we explore a wider range of vortex extremes, thereby broadening the results in H&M19 concerning the importance of interactive chemistry for simulating stratospheric vortex variability and stratosphere-troposphere coupling.

$<<$

**General comments: This is essentially a comparison between two different models, one that has been exhaustively tuned at NCAR to provide the best possible climatology and variability (CHEM), and one that has been "downgraded" by**

**specifying the evolution of ozone and other species (NOCHEM).**

$>>$

We fear the reviewer's view of SC-WACCM is a misconception. The WACCM Specified Chemistry (SC-WACCM) model is officially supported and scientifically validated by NCAR and it can be run using prescribed ozone in the same fashion as other models used for CMIP6 (Keeble et al., 2020). Under preindustrial conditions, it was thoughtly tested, and the stratospheric dynamics and stratosphere-troposphere coupling in WACCM and SC-WACCM were shown to be very similar across a wide variety of metrics (Smith et al., 2014). These features make it appropriate to pair these two to study the contributions in a model of interactive chemistry and transport. There is nothing "downgraded" in SC-WACCM: the ozone specification is SC-WACCM is identical to the one implemented by the vast majority of CMIP5 and CMIP6 models. Hence, it is of great interest to document how WACCM and SC-WACCM differ, in order to understand how interactive ozone chemistry affects climate variability.

$<<$

**Line 18. Interannual variability of what?**

**First paragraph of section 3.1. It should be good to at least speculate about a mechanism to explain the stronger westerlies in CHEM. In the light of the given comparisona against Haase and Matthes and Smith et al, are high CFC concentrations needed to get this result?**

**Line 168-169. "(...) perhaps because of greater wave activity necessary for an SSW to occur with a stronger mean vortex state (...)". The authors provide no proof of this,but it should be easy to check.**

**Figure 6. Please consider removing this figure, it does not really add much to**

the manuscript.

**Conclusions, first sentence. This has not beens shown, at least to my under-standing. A mechanism behind the circulation differences between CHEM and NOCHEM has not been provided, nor have the authors discriminated between the effects of ozone chemistry and the effects of transport of ozone during SSWs.**

**Line 253-254. It is a speculation that CHEM needs stronger wave forcing to drive SSWs, it has not been proven. Please modify this sentence accordingly.**

**Please refer to individual figure panels using the labels a, b, c, etc. and not top-right etc.**

$>>$

We have already addressed all these comments in response to Referees 1 or 2. See above. In short, we added results for a wave forcing diagnostic (eddy heat flux at 100 hPa), and also explored changes in downwelling velocities.

We also tested the robustness of this result (stronger vortex in CHEM than NOCHEM) by comparing the vortex strength in the set of transient historical simulations documented in Neely et al. (2014). These are 10 ensembles with interactive (WACCM) or specified ozone (SC WACCM), as monthly and daily climatologies. Again, we found the same results, suggesting a stronger vortex with interactive ozone in late-winter/spring, indicating that this effect is robust. Nonetheless, the details of the relationship to CFCs will be investigated in a follow-up study using more models. We have added a brief discussion of this in the text where we discuss the differing vortex climatologies in CHEM and NOCHEM, i.e. Lines 81-85 and 122-127.

$<<$

**Line 119. I find it surprising that a 1.7 m/s difference be statistically significant, given the large interannual variability in the strength of the westerlies. But letting aside the statistics, how physically meaningful might this difference be?**

$>>$

We have added further detail on the statistical testing here (see response to Referee 2). In terms of physical meaning, this difference seems to be related to a final warming date that is on average a week later (a large difference compared to variability) and large differences in numbers of SPVs and late winter SSWs (through the later final warming date). Figure 7 also shows a stronger vortex leading up to a midwinter SSW central date in CHEM than in NOCHEM, which is related to the strength of the SSW itself. So this mean state difference, while small, does seem to have physical meaning.

$<<$

**In line 153 it is stated that Fig. 3 reveals the surface impacts of SSWs. In the beginning of the following paragrah it is addressed whether the results of Fig.3 are due to SSWs. Please consider presenting first Fig. 4, and then Fig. 3 (or revise the way the story line is presented).**

$>>$

We agree with the reviewer that this was unclear. We have reworded for a more coherent storyline.

Line 171-172: "To determine whether the differences at the surface following SSWs in CHEM and NOCHEM are a result of differences in the events in the stratosphere, we calculate the Northern Annular Mode (NAM) for each simulation."

$<<$

**Line 163. I am not so sure about this. The positive T signal in Fig. 5a appears at**

**positive lags, generally after the peak of the warming. Without further analysis, this might well be interpreted as a slower temperature recovery in the aftermath of SSWs in CHEM than in NOCHEM.**

$>>$

While the bulk of the positive temperature signal is at positive lags, it does appear at negative lags too; we have added a better marker of the central date (0 lag) to make this clearer in Fig. 5.

$<<$

**An alternative explanation of stronger dynamical heating in CHEM could be that having a stronger jet in the mid-stratosphere generally implies a stronger vertical gradient of temperature over the pole (see Fig. 3b In Haase and Matthes). And a given w\* anomaly, under a stronger temperature vertical gradient, would produce a stronger dynamical heating.**

$>>$

We thank the referee for their insightful comment. We calculated $\overline{w}^*$ using daily data but found no difference between CHEM and NOCHEM near the onset of SSWs, so the alternate explanation here (larger vertical temperature gradient) seems a plausible part of the dynamical mechanism. However, the details are still unclear and will be the subject of follow-up work involving more models. This is discussed on Lines 189-197.

Lines 189-197: "The increased dynamical heating in CHEM could be related to greater wave activity necessary for an SSW to occur with a stronger mean vortex state. Figure 6 shows the eddy heat flux over 40-80° N over time in CHEM and NOCHEM. This is stronger by about 2 mK/s just before the central date in CHEM than in NOCHEM, indicating stronger wave forcing in CHEM. The CHEM and NOCHEM means are at the upper and lower bounds of the other's confidence intervals, respectively. Further, the zonal mean zonal winds at 10 hPa and 60° N around the central date of the SSW

(shown in Figure 7) are both stronger prior to the event and more easterly following the central date in CHEM than in NOCHEM. However, the residual vertical velocity (not shown) is similar leading up to SSWs for CHEM and NOCHEM, so it is possible that the increased dynamical heating is a result of a stronger vertical temperature gradient related to the stronger vortex (associated with a colder pole)."

$<<$

**Figure 5. It would be good to include the temperature tendency to better understand the effects and timing of the dynamical heating (please be explicit, is this w*dt/dz, TEM formalism?), and the radiative heating. Also, please include in the caption the latitude band considered.**

$>>$

This is dynamical heating as model output, which represents the temperature tendency computed by the dynamical core. Hence, it is not the TEM formalism, but it should be fairly close to the TEM approximation (w*dT/dz). As indicated above, the dynamical heating following the TEM formalism will be more thoroughly studied in follow-up work involving more models. Lastly, we have added the latitude band ($60 - 90°$ N) to the caption as suggested.

$<<$

**Line 170-172. So would it be possible to discriminate the fraction of the stronger long-wave cooling that comes from having reached higher temperatures, from the fraction that comes from having larger ozone concentrations? This would tell us the importance of the accumulation of ozone over the pole during SSWs to the recovery of the vortex.**

$>>$

This is an interesting question, but it would require running the radiation scheme offline to fully account for the ozone impacts on the LW. This is unfortunately unfeasible at present, but it could be studied in the future.

<<

**Lines 183-190. The study of De La Camara et al also shows results on changes of polar ozone during SSWs using a similar version of WACCM as the one used here.**

>>

We have reworded the references to this study to acknowledge both the reanalysis and model-based results reported in De La Camara et al. See Lines 205-207 and 209-210.

Lines 205-207: "We see a sharp increase in ozone in the 15 days leading up to the central date, reaching a peak of on average about 40 Dobson units above climatology just after the central date, similar to that seen in reanalysis and a similar model by De La Cámara et al. (2018)."

Lines 209-210: "This ozone anomaly is consistent with total ozone column in reanalysis and a similar model (De La Cámara et al., 2018) and the smaller ozone depletion in years with early SSWs observed by Strahan et al. (2016)."

<<

**Line 200. Please take into account that NAM decorrelation timescales are shorter in March-April than in winter (Simpson et al. 2011), which may help understand the weaker persistence of surface anomalies after March SSWs.**

>>

We thank the reviewer for noting this; we have added it to the discussion. See Lines

218-220.

Lines 218-220: "Three factors could contribute to this: weakerSSWs, weaker stratosphere-troposphere coupling, both discussed below, and a shorter NAM decorrelation timescale in March than in DJF (Baldwin et al., 2003; Simpson et al., 2011), which would result in weaker anomalies at the surface when averaged over several weeks."

<<

**I am unsure whether it is worth keeping section 3.4, but 4 figures to only show a very weak signal seems excessive.**

>>

We have moved two of the four figures to the supplementary material. However, as stated above, we believe that the insignificant difference here is worth documenting and discussing.

Please also note the supplement to this comment:
https://www.atmos-chem-phys-discuss.net/acp-2020-187/acp-2020-187-AC1-supplement.pdf

**Supplement:**

[revised manuscript text omitted]

As in Figure 4, for March SSWs.

As in Figure 5, for March SSWs.

[Figure]

**Figure 9.** As in Figure 7, for March SSWs.

[Figure]

**Figure 10.** As in Figure 3, for DJF SPVs.

As in Figure 4, for DJF SPVs.

475    As in Figure 5, for DJF SPVs.

[Figure]

**Figure 11.** As in Figure 7, for DJF SPVs.

[Figure]

**Figure S1.** CHEM-NOCHEM difference in shortwave heating anomalies from -30 to +60 days around the SSW central dates in December (a), January (b), and February (c).

[Figure]

**Figure S2.** As in Figure 4, for March SSWs.

[Figure]

**Figure S3.** As in Figure 5, for March SSWs.

[Figure]

**Figure S4.** As in Figure 6, for March SSWs.

[Figure]

**Figure S5.** As in Figure 4, for DJF SPVs.

[Figure]

**Figure S6.** As in Figure 5, for DJF SPVs.

---

## Author Response (AR2)

Dear Dr. Grooß,

We thank you for your comments and have made the suggested minor revisions. Those revisions and detailed replies to comments are below.

<<

**Reviewer 2: "I think the mechanisms for the differences they are seeing could be better explained..."**

**Reviewer 3: "The paper also lacks a clear explanation of the mechanisms behind the differences in basic state and variability between CHEM and NOCHEM..."**

**In the conclusions you speculate about a stronger wave forcing. But could you say more about the mechanisms behind this coupling?**

>>

In context, Reviewer 2 here seems to be referring to the differences following SSWs. This is the mechanism related to stronger wave forcing, which we discuss on page 7 and provide evidence of in Figures 5-7. We have added an additional paragraph to further clarify and summarize this mechanism. To the extent that Reviewer 2 is commenting on differences in the basic state of the polar stratosphere, we address this along with Reviewer 3's comments below.

Lines 223-225: "In summary, DJF SSWs are preceded by larger wave forcing in CHEM than in NOCHEM, partially because of the stronger mean state of the polar vortex. This then results, on average, in more intense SSWs, stronger stratosphere-troposphere coupling, a more negative NAO-like pattern at the surface, and long-lasting stratospheric ozone anomalies."

On the subject of adding more details about a mechanism for the difference in the basic state of the polar stratosphere, as Reviewer 3 asks: we have added a long paragraph on page 5 concerning a possible mechanism for the stronger polar vortex when interactive ozone is included. More specifically, ozone asymmetries may play a role by pre-conditioning the waves, leading to a colder polar vortex (Alber and Nathan, 2012). However, we think referee 3 is a little too optimistic that a robust mechanism can indeed be determined. The system we are dealing with is highly non-linear, as sudden warmings are large-amplitude breaking waves, so most linear approximations would not apply. In addition, the coupling with ozone chemistry and transport adds yet another level of complexity, rendering the modeled system very complicated and chaotic (note that extreme vortex events, such as SSWs and SPVs, are partly the result of such chaotic variability). We also note the previous paper on this subject using a model of similar complexity (Haase and Matthes 2019), also in ACP, which our paper confirms and extends on in a number of ways, make no attempt to establish a mechanism for the changes in the basic state either, for precisely the same reason. Hence, we do not think our study suffers from a major flaw, and we would rather stick to the facts than speculate too extensively on potential mechanisms that may or may not be applicable. We have nonetheless added an explanation about a possible mechanism and thank the editor referee for their comment.

Lines 129-139: "Because the differences between interactive and specified ozonesimulations depend on the level of CFCs, a precise understanding of the mechanisms for the difference will require disentangling the dynamics and chemistry. Higher ozone variability in the presence of CFCs (Calvo et al., 2015) might increase the effects of the ozone-dynamics feedbacks, rendering this a very difficult problem. There are indications that these differences may be related to zonal asymmetry of ozone (Haase and Matthes, 2019), further complicating the relationship. Albers and Nathan (2012) have proposed a complex mechanism to detail the coupling of zonally asymmet-ric ozone and dynamics in the context of a highly idealized linear model. In their model, zonal asymmetries in ozone precondition the waves, causing a reduction in planetary wave drag and a colder polar vortex. However, determining whether this mechanism is operative in our comprehensive model would be quite difficult, as the mechanism relies on many assumptions that are likely inapplicable in the presence of highly nonlinear, time-dependent, breaking waves as are observed in the winter polar stratosphere in a fully interactive model."

<<

**Introduction: Since you give the definition of SSWs and SPVs later, in section 2 section, it maybe easier for the reader to add a very short explanation here or at least a hint that these are defined in section 2.**

>>

We have added some additional description of these events (SPVs in particular) in the introduction and have stated that they will be fully defined in Section 2.

Lines 35-36: "We define these precisely in Section 2, based on extreme values of zonal mean zonal wind."

Lines 39-40: "Conversely, SPVs, in which abnormally strong westerly zonal mean zonal winds occur, are the result of anomalously weak planetary wave activity over a protracted period."

<<

**Data availability: please give some detail how to access the data (like URL, simulation abbreviation ...)**
>>

We have added in the simulation abbreviations in the data availability section.

<<

**Supplement: please include also a short text paragraph of what is shown in the figures. A formal title page of the supplement will be added by ACP. It may be helpful to also add the day=0 line also in these plots.** >>

We have added a short introductory paragraph, expanded the captions, and added day=0 lines to all temperature and heating evolution plots (figures that are parallel to those with day=0 lines in the main text).

Supplement Lines 1-6: "In this supplementary section, we show the seasonality of the shortwave heating anomalies induced by the CHEM (interactive chemistry) simulation compared to NOCHEM (specified chemistry) in the occurrence of SSWs (Fig. S1). Given that March SSWs behave differently from those occurring in midwinter, we show the evolution of these events separately from that of the DJF SSWs. In particular, here we show the NAM evolution (Fig. S2), evolution of temperature and the individual heating terms (Fig. S3), and the wave forcing (Fig. S4). (These figures are parallel to Figures 4, 5, and 6 for midwinter SSWs in the main text). Finally, we show the same sequence (NAM, temperature and heating terms) for SPVs in midwinter in Figs. S5-S6."

[revised manuscript text omitted]

In this supplementary section, we show the seasonality of the shortwave heating anomalies induced by the CHEM (interactive chemistry) simulation compared to NOCHEM (specified chemistry) in the occurrence of SSWs (Fig. S1). Given that March SSWs behave differently from those occurring in midwinter, we show the evolution of these events separately from that of the DJF SSWs. In particular, here we show the NAM evolution (Fig. S2), evolution of temperature and the individual heating terms (Fig. S3), and the wave forcing (Fig. S4). (These figures are parallel to Figures 4, 5, and 6 for midwinter SSWs in the main text). Finally, we show the same sequence (NAM, temperature and heating terms) for SPVs in midwinter in Figs. S5-S6.

[Figure]

**Figure S1.** CHEM-NOCHEM difference in shortwave heating anomalies from -30 to +60 days around the SSW central dates in December (a), January (b), and February (c).

[Figure]

**Figure S2.**  NAM anomaly composites around March SSW central dates in CHEM (a), NOCHEM (b), CHEM-NOCHEM (c). Stippling shows significance at the 95% level (with a Monte Carlo test for CHEM and NOCHEM and a two-tailed t-test for CHEM-NOCHEM). Contours are every 0.5 standard units for CHEM and NOCHEM and every 0.2 standard units for CHEM-NOCHEM.

[Figure]

**Figure S3.**  CHEM-NOCHEM differences   the temperature and heating anomalies over 60-90° N from -30 to +60 days around the SSW   central dates. (a): Temperature anomalies. Contours are every 1 K. (b): Dynamical heating anomalies. Contours are every 0.5 K/day. (c): Longwave heating anomalies. Contours are every 0.25 K/day. (d): Shortwave heating anomalies. Contours are every 0.02 K/day. Stippling shows significance at the 95% level under a two-tailed t-test.

[Figure]

**Figure S4.**  Eddy heat flux in  mK/s over 40-80° N from -60 to +30 days around the SSW March  central dates. The CHEM average is in blue, with confidence intervals shown in pale blue. The NOCHEM average is in black, with confidence intervals shown in gray.

[Figure]

**Figure S5.**  NAM anomaly composites around DJF SPV central dates in  CHEM (a), NOCHEM (b), CHEM-NOCHEM (c). Stippling shows significance at the 95% level (with a Monte Carlo test for  CHEM and NOCHEM and a two-tailed t-test for CHEM-NOCHEM). Contours are every 0.5 standard units for CHEM and NOCHEM and every 0.2 standard units for CHEM-NOCHEM.

[Figure]

**Figure S6.**  CHEM-NOCHEM differences  the temperature and heating anomalies over 60-90° N from -30 to +60 days around the SPV DJF  central dates. (a): Temperature anomalies. Contours are every 1 K. (b): Dynamical heating anomalies. Contours are every 0.5 K/day. (c): Longwave heating anomalies. Contours are every 0.25 K/day. (d): Shortwave heating anomalies. Contours are every 0.02 K/day. Stippling shows significance at the 95% level under a two-tailed t-test.